# Grave-to-cradle upcycling of Ni from electroplating wastewater to photothermal $CO_2$ catalysis

Shenghua Wang[1,8], Dake Zhang[1,2,8], Wu Wang[3], Jun Zhong[2], Kai Feng[4], Zhiyi Wu[2], Boyu Du[5], Jiaqing He[3], Zhengwen Li[4], Le He [2,6] ✉, Wei Sun [1] ✉, Deren Yang[1] ✉ & Geoffrey A. Ozin [7] ✉

Treating hazardous waste Ni from the electroplating industry is mandated world-wide, is exceptionally expensive, and carries a very high $CO_2$ footprint. Rather than regarding Ni as a disposable waste, the chemicals and petro-chemicals industries could instead consider it a huge resource. In the work described herein, we present a strategy for upcycling waste Ni from electro-plating wastewater into a photothermal catalyst for converting $CO_2$ to CO. Specifically, magnetic nanoparticles encapsulated in amine functionalized porous $SiO_2$, is demonstrated to efficiently scavenge Ni from electroplating wastewater for utilization in photothermal $CO_2$ catalysis. The core-shell cata-lyst architecture produces CO at a rate of 1.9 $mol \cdot g_{Ni}^{-1} \cdot h^{-1}$ (44.1 $mmol \cdot g_{cat}^{-1} \cdot h^{-1}$), a selectivity close to 100%, and notable long-term stability. This strategy of upcycling metal waste into functional, catalytic materials offers a multi-pronged approach for clean and renewable energy technologies.

The annual discharge of heavy metal wastewater produced by the electroplating industry has exceeded 4 billion tons just in China[1]. Ni electroplating is particularly widely practiced and allowed to enter natural water systems, causing serious lung, kidney, stomach, intes-tine, and skin diseases in humans[2]. While Ni is less costly than precious metals, it has more health and safety regulatory standards by autho-rities like the EPA[3-5]. The discharged Ni has been commonly regarded as waste or hazard that needs to be treated via remediation methods, like chemical precipitation and electrodeposition[6,7]. They require preconcentration, post-treatment of sludge, and electricity input, which are often accompanied by high energy costs and a large $CO_2$ footprint[8,9]. In the meantime, other industries like chemicals and pet-rochemicals, as well as electroplating itself that requires an excessive amount of Ni are stimulating mining of more Ni from minerals and

paying the added price of reagent-grade Ni chemicals, leaving the waste Ni from electroplating buried underground. This bleak situation is worrisome in the climate-challenged world, where the greenhouse effect is looming, and the transition towards sustainable energy and a green economy is of great urgency[10-16]. Fortunately, emerging waste-water techniques, including adsorption[5,17-19] and ion exchange[20], many coupled with magnetic separation[21-24], provide energy-saving solutions to scavenge and recycle Ni from the wastewater, but the big question is: what happens next to the adsorbents and exchangers? Landfilling them causes further environmental and health issues, whereas regen-erating them and retrieving Ni costs more reagents and energy.

Richard Buckminster-Fuller in Life Magazine said, "Pollution is nothing but resources we're not harvesting". Today we are not unfa-miliar with the concept of "recycling" waste, but industries' bottom

[1]State Key Laboratory of Silicon Materials, School of Materials Science and Engineering, Zhejiang University, 310027 Hangzhou, Zhejiang, P. R. China. [2]Institute of Functional Nano & Soft Materials (FUNSOM), Soochow University, 215123 Suzhou, P. R. China. [3]Shenzhen Key Laboratory of Thermoelectric Materials, Department of Physics, Southern University of Science and Technology, 518055 Shenzhen, Guangdong, China. [4]Department of Chemical Engineering, Tsinghua University, 100084 Beijing, China. [5]Changchun Ecological Environment Monitoring Center, Changchun, Jilin Province, China. [6]Jiangsu Key Laboratory of Advanced Negative Carbon Technologies, Soochow University, 215123 Suzhou, Jiangsu, P. R. China. [7]Materials Chemistry and Nanochemistry Research Group, Solar Fuels Cluster, Departments of Chemistry, University of Toronto, Toronto, ON M5S 3H6, Canada. [8]These authors contributed equally: Shenghua Wang, Dake Zhang. ✉e-mail: lehe@suda.edu.cn; sunnyway423@zju.edu.cn; mseyang@zju.edu.cn; gozin@chem.utoronto.ca

lines often demand that the resulting product of recycling must be better or even higher value than the original item. This paradigm of so-called "upcycling" has become a keyword in the new world of technologies being eco-friendly[25,26]. In this context, an economically and environmentally viable strategy should be to directly transform the scavenged Ni from electroplating wastewater to a valuable product that avoids energy- and $CO_2$-intensive mining and rather stores energy in chemicals and fuels made from $CO_2$. Although nickel has been widely accepted as an excellent catalyst for $CO_2$ reduction[27–30], this vision is challenging because electroplating wastewater has a complex composition containing Ni salts with multiple anions, including sulfate, chloride, and amino-sulphonate, which may hinder the recovery of pure Ni chemical feedstocks, and even if an adsorbent is used to retrieve Ni regardless of the anion mix, the adsorbing components can be redundant or even detrimental to the new product. For example, Mikhail et al.'s work suggests that the presence of Na and K impurities in $Ni/CeZrO_x$ catalysts resulted in decreased $CO_2$ conversions, lower selectivity to $CH_4$, and increased power consumption in DBD plasma-catalytic $CO_2$ methanation[31]. In contrast, the impurities can also function as promoters rather than inhibitors. In many cases, different from Maria's case, the introduction of alkali metals (e.g., Na and K) could instead improve the catalytic performance[32–34]. Nevertheless, the existence of impurities in wastewater still brings complications and uncertainties, so the judicious design of the upcycling process, materials, and catalysts are critical and must be validated.

Targeting this challenge, we demonstrate a two-fold direct recycling and upcycling strategy to scavenge the majority of the Ni from zero-cost and abundant electroplating wastewater and utilize it as a heterogeneous photothermal catalyst for converting $CO_2$ to chemicals and fuels. A carefully designed adsorbent comprised of a sphere-shaped, multifunctional ternary heterostructure $Fe_3O_4@SiO_2@mSiO_2$-$NH_2$ was utilized to capture high dispersions of Ni in real electroplating tailings, employing the chemical tethering property of $-NH_2$ enhanced by the large surface area of mesopores in $SiO_2$. The ternary nano-composite $Fe_3O_4@SiO_2@mSiO_2$-$NH_2@Ni$ is comprised of earth-abundant elements. It displays a remarkable 100% selectivity and high stability towards the reverse water-gas shift reaction $CO_2 + H_2 \rightarrow CO + H_2O$, qualities not enjoyed by traditional Ni-sourced heterogeneous catalysts. The Fe core provides additional benefits that include straightforward magnetic separation of the $Fe_3O_4@$-$SiO_2@mSiO_2$-$NH_2@Ni$ catalyst from the electroplating wastewater, a light-activated broadband nano heater to drive the photothermal reverse water-gas shift reaction at higher efficiency, milder conditions, and a lower $CO_2$ footprint than the thermochemical reverse water-gas shift reaction.

## Results

Our strategy to upcycle Ni from real electroplating wastewater can be divided into several steps (Fig. 1). First, a designed adsorbent is suspended in real Ni(II)-containing electroplating wastewater (step 1). After reaching adsorption equilibrium, the adsorbed Ni(II), together with the adsorbent, is collected via a magnet (step 2) and then directly calcined to be transformed into an active catalyst with a nanoreactor-type architecture (step 3). The obtained catalyst is then used in (photothermal) catalytic $CO_2$ hydrogenation reactions (step 4). The architecture of the nanoreactor catalyst was carefully designed to realize the following functionalities: the Fe core of our nanoreactor supplies sustainable photothermal heat, and the Ni within the silicon oxide synthesizes solar fuel.

### Characterization of the catalysts

The adsorbent used in this work was designed to have a core-mantle-crust structure, with a magnetic core ($Fe_3O_4$ colloidal nanocrystal clusters, denoted as CNC) and two layers of silica (Supplementary Fig. 1), employing the first and second most abundant element in the

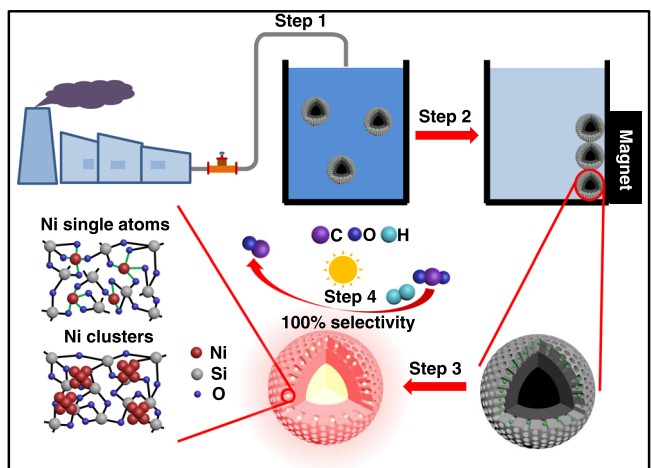

**Fig. 1 | Schematic demonstrating the recycle of Ni from real electroplating wastewater.** The Ni resources in the electroplating wastewater were upcycled for photothermal $CO_2$ catalysis applications.

Earth's crust: O and Si. The strong magnetism of the CNC core ensured easy separation of adsorbents from solution[35–37]. The dense silica in the middle mantle layer prevented the CNC core from being etched in the acidic wastewater. The outmost crust of mesoporous silica served to enlarge the specific surface area (SSA) to benefit adsorption and further catalytic reaction. Besides, the adsorbent particles (denoted as $CNC@SiO_2@mSiO_2$) were also grafted with amino groups (denoted as $S_{ad}$) to facilitate the capture of Ni(II). The Brunauer−Emmett−Teller (BET) surface areas of the amino-grafted adsorbents with and without the mesoporous silica layer were calculated to be 301.7 and 16.3 $m^2 g^{-1}$, respectively, demonstrating the function of mesoporous silica in increasing SSA (Supplementary Fig. 2). A sample prepared by coating CNC with only a mesoporous $SiO_2$ layer (without the middle mantle protecting layer, denoted as $CNC@mSiO_2$) was used as a control sample to verify the acid resistance of the dense mantle $SiO_2$ layer in $CNC@SiO_2@mSiO_2$ (Supplementary Fig. 3). The similar particle size distributions of $CNC@SiO_2@mSiO_2$ and $CNC@mSiO_2$ suggest that the thickness of $SiO_2$ for these two samples are the same, making the comparison fair. The concentration of Fe in the etching solution for $CNC@SiO_2@mSiO_2$ is much lower than that for $CNC@mSiO_2$ under various etching conditions. These results indicate that the dense silica in the middle layer of $CNC@SiO_2@mSiO_2$ has strong resistance against acid.

After the treatment of the electroplating wastewater, the recycled $Ni^{2+}$ together with the adsorbents was directly calcined in air and $H_2$ sequentially (denoted as $S_{Fe-Ni}$), which completed the assembly of the nanoreactor catalyst. Figure 2 shows the morphology and composition of the obtained catalyst. The overall core-shell structure was retained, while the CNC core was cracked (Fig. 2b and Supplementary Fig. 1). Notably, no apparent Ni nanoparticles could be observed, suggesting that Ni should be highly dispersed in $S_{Fe-Ni}$. This was further corroborated by the X-ray diffraction (XRD) results (Supplementary Fig. 4). The patterns for $S_{Fe-Ni}$ showed no characteristic peaks of Ni crystals, excluding the generation of large particles and aggregates. Whereas, the characteristic peaks of Fe were clearly observed in the patterns, suggesting that the CNC core had been reduced in the calcination process. The high dispersion of Ni in $S_{Fe-Ni}$ was ultimately confirmed by the aberration-corrected high-angle annular darkfield-scanning transmission electron microscopy (HAADF-STEM) images (Fig. 2c), in which no nanoparticles but only bright dots could be observed. Energy Dispersive Spectrometer (EDS) mappings further demonstrated the core-mantle-crust structure ($CNC@SiO_2@mSiO_2$) in which Ni was successfully loaded (Fig. 2d–h).

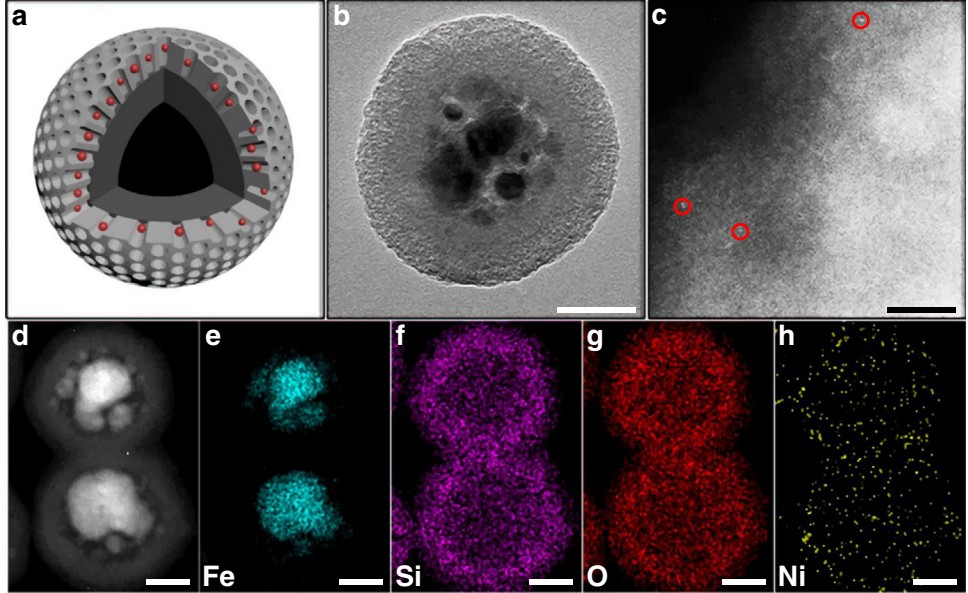

**Fig. 2 | Morphology and components of the carefully designed S_{Fe-Ni} catalyst. a** Schematic of S_{Fe-Ni}. The black core, the dark gray middle layer, the light gray porous outer layer and the red spheres represent CNC, the dense silica, the mesoporous silica, and Ni, respectively. **b** TEM image of S_{Fe-Ni}. Scale bar, 50 nm. **c** HAADF-STEM image of S_{Fe-Ni}. Scale bar, 2 nm. **d–h** EDS mappings of S_{Fe-Ni}. Scale bars, 50 nm.

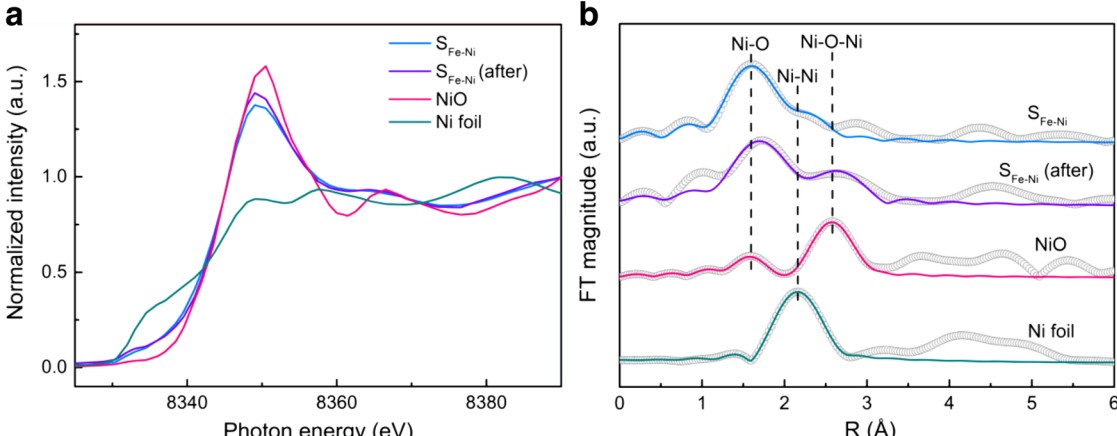

**Fig. 3 | XAS fine structures of the fresh and the used S_{Fe-Ni} catalysts. a** Ni K-edge XANES spectra of the fresh and used S_{Fe-Ni} catalyst, exhibiting the normalized absorption intensity as a function of the X-ray photon energy. The used catalyst is denoted as S_{Fe-Ni} (after). Ni foil and NiO were used as references. **b** Ni K-edge EXAFS fitting results of the fresh and used S_{Fe-Ni} catalyst. The black open circles show the original data of the EXAFS spectra, while the lines of other colors are the fitting results of the spectra.

To identify the chemical environment in the sub-nano dimension and the electronic structure of Ni in S_{Fe-Ni}, X-ray absorption near-edge structure (XANES) and extended X-ray absorption fine structure (EXAFS) spectra at the Ni K-edge for S_{Fe-Ni} were obtained (Fig. 3 and Supplementary Table 1). In the Ni K-edge XANES spectrum, the near-edge absorption energy of S_{Fe-Ni} is located between the Ni foil and NiO, indicating the partially oxidized state of Ni in S_{Fe-Ni}. The FT-EXAFS curve for S_{Fe-Ni} shows a main peak at ~1.6 Å, which could be attributed to Ni–O scattering[38]. A very small peak attributed to the Ni–Ni scattering signal (located at ~2.1 Å) for S_{Fe-Ni} demonstrates the existence of aggregated Ni is rare[38]. Fitting the peak of the first shell of Ni–O shows the average coordination number (CN) of oxygen atoms around a Ni atom (CN (Ni–O)) in S_{Fe-Ni} is 3.8, indicating that Ni is unconventionally coordinated in comparison to NiO, which has a CN (Ni–O) of 6[39]. The decreased CN (Ni–O) of S_{Fe-Ni} should be ascribed to the highly dispersed adsorbed state of Ni on the surface of SiO_2 without NiO nanoparticles. The second coordinative shell can be assigned to Ni–Ni coordination, of which the CN (Ni–Ni) is 4.8. The much lower CN (Ni–Ni) of S_{Fe-Ni} than that of Ni foil (CN (Ni–Ni) = 12)[39], combined with the aberration-corrected HAADF-STEM results, suggests that our catalyst preparation strategy ensured that Ni existed as highly dispersed atoms and ultrasmall clusters, rather than aggregated nanoparticles.

As reported, small-sized Ni catalysts exhibit excellent catalytic performance in CO_2 hydrogenation[40]. Therefore, the unique structure of S_{Fe-Ni} should render it a potential candidate for this reaction. Meanwhile, the Fe core, as an active metal and a great photothermal material, could offer additional activity to the system. Therefore, we conducted systematic catalytic tests to evaluate the catalytic performance of this designed catalyst and identify the roles of the key components, under both thermal and photothermal conditions. For comparison, two primary control samples excluding either Fe or Ni were investigated to manifest the rationality of our catalyst design,

under the same testing conditions: a control sample prepared by annealing the bare $S_{ad}$ without Ni under the same condition as $S_{Fe-Ni}$ (denoted as $S_{Fe}$, Supplementary Fig. 5), and another control sample substituting the CNC core with a $SiO_2$ core (denoted as $S_{Ni}$) by treating the $SiO_2@mSiO_2$ nanoparticles with the same amino functionalization, adsorption, and annealing procedure of $S_{Fe-Ni}$ (Supplementary Fig. 6). $S_{Fe}$ exhibited the same morphology of $S_{Fe-Ni}$, while no Ni could be found from the EDX element maps. Meanwhile, a solid silica core was found for $S_{Ni}$ with Ni uniformly dispersed in the outer silica layer, and no Fe was observed in the EDX element maps.

## Thermocatalytic performance

Before pursuing the sustainable potential in light-driven $CO_2$ reduction, we first evaluated the traditional thermocatalytic performances of the samples to provide fundamental insights into the catalytic sites, product selectivity, and thermal stability of our catalyst. It was found that the CO production rates of $S_{Fe-Ni}$ were much higher than those of $S_{Fe}$ at all testing temperatures (Supplementary Fig. 7a). Below 400 °C, the control sample $S_{Fe}$ hardly even showed observable activity. This indicates that the loading of Ni led to pronounced improvement in catalytic performance for $S_{ad}$. Notably, the CO rate of $S_{Ni}$ was close to that of $S_{Fe-Ni}$. This further demonstrates that the catalytic ability of our designed architecture under pure thermal conditions mainly comes from Ni rather than the Fe core. Only at the relatively high temperatures of 450 and 500 °C, when the diffusion of reactants was greatly enhanced, the Fe component could start to contribute to the yield of CO, indicated by the increasingly observable rate of $S_{Fe}$ and the higher rate of $S_{Fe-Ni}$ than $S_{Ni}$ at these temperatures. Thanks to the sub-nano size of Ni−O species, only a trace amount of $CH_4$ was found in the products (Supplementary Fig. 8), consistent with the literature that ultrasmall Ni if oxidized, would favor the production of CO from hydrogenation of $CO_2$[38,40,41]. Nearly 100% of CO selectivity (99.998%) was thus achieved for $S_{Fe-Ni}$ at 500 °C, which is in contrast with traditional supported Ni nanoparticles that often yield a significant proportion of $CH_4$ and would necessitate an energy-intensive separation process for the products. For example, the selectivity of $CH_4$ was found to be 58.6% for commercially available silica-alumina supported Ni nanoparticles ($Ni/SiO_2·Al_2O_3$) tested in our reactor (Supplementary Table 2).

To examine whether the mesopores contribute to the small size of Ni and, in turn, the great selectivity, another control sample was prepared by using solid $SiO_2$ spheres as the support to graft amino groups, followed by the adsorption and activation of Ni similar to other samples (denoted as $S_{SiO2-Ni}$). As Supplementary Fig. 9a-b shows, no large Ni particles, and no obvious crystalline Ni signals could be observed from the TEM and XRD results, respectively. Moreover, the thermocatalytic results suggest that CO was the only product for $S_{SiO2-Ni}$, which is characteristic of small-sized Ni (Supplementary Fig. 9c). The BET surface area of amino-grafted solid $SiO_2$ ($S_{SiO2-NH2}$) is much lower than that of $CNC@SiO_2@mSiO_2-NH_2$ (15.9 $m^2/g$ vs. 301.7 $m^2/g$, Supplementary Fig. 9d). This leads to the much poorer catalytic activity of $S_{SiO2-Ni}$, only half of that of $S_{Fe-Ni}$. These results indicate that the mesoporous $SiO_2$ only act as effective support with a high specific surface area.

As Supplementary Table 3 shows, many other ions besides $Ni^{2+}$ exist in the electroplating wastewater. To verify the influence of these ions on the catalytic performance, a synthetic $Ni^{2+}$ solution made from reagent $Ni(NO_3)_2·6H_2O$ with only $Ni^{2+}$ (the $Ni^{2+}$ concentration is the same as that for the real electroplating wastewater) was used as the precursor to prepare the $S_{Fe-Ni}$ sample. Notably, Supplementary Fig. 10 shows that the CO rate for $S_{Fe-Ni}$ prepared from real electroplating wastewater is the same as that for $S_{Fe-Ni}$ prepared from the synthetic $Ni^{2+}$ solution at 400 °C, while only a slightly lower CO rate was found for the former than for the latter at 500 °C. Nevertheless, the overall

discrepancy between these two $S_{Fe-Ni}$ samples is minimal. This should be attributed to the similar loading amount of Ni for these two samples and the trace amount of other ions therein (Supplementary Tables 4 and 5). These results suggest that the existence of the other ions besides $Ni^{2+}$ in the real electroplating wastewater does not deteriorate the upcycling efficiency of Ni for the $CO_2$ catalysis reported herein. Overall, the above results preliminarily manifest that our designed adsorbent could successfully recycle Ni from the electroplating wastewater into a highly active and selective catalyst for $CO_2$ hydrogenation reactions.

The stability of $S_{Fe-Ni}$ was evaluated through testing at the relatively high temperature of 500 °C, shown in Supplementary Fig. 7b. After a slight decrease, the CO rate began to increase slowly and tended to be stabilized above 5 mmol $g_{cat}^{-1}$ $h^{-1}$. The mild decrease of yield in the initial stage should arise from the oxidation of Ni, demonstrated by the XAS results (Fig. 3 and Supplementary Table 1). The used catalyst, denoted as $S_{Fe-Ni}$ (after), showed a very similar Ni K-edge XANES spectrum with $S_{Fe-Ni}$, but the pattern became slightly closer to that of NiO (Fig. 3a). In the Ni K-edge EXAFS spectrum, a small Ni−O-Ni hump (located at ~2.6 Å) emerged while the small peak for Ni−Ni disappeared[41]. This indicates that the Ni clusters had been oxidized after the reaction. Meanwhile, the intensity of the Ni−O with low CN was maintained after the stability test, which is in line with the preserved high catalytic activity and selectivity towards CO production. Notably, more Ni atoms and clusters without serious aggregation could be seen in the used catalyst (Supplementary Fig. 11c) than in the fresh samples (Fig. 2c). The high dispersion of Ni in the used samples could also be evidenced by the XRD patterns which show no characteristic signals of crystalline Ni (Supplementary Fig. 12). The migration of Ni from the inner part of the mesopores to the more outer part was beneficial for the mass transfer of the reactants, leading to the gradual increase of production rate seen in Supplementary Fig 7b.

The catalytic pathway was studied through the in situ diffuse reflectance infrared Fourier transform spectroscopy (DRIFTS) experiments and kinetic studies (Supplementary Figs. 13–15). The in situ DRIFTS spectra of $S_{Fe-Ni}$ (Supplementary Fig. 13a) show weak signals of gaseous CO (2111 and 2174 $cm^{-1}$)[28,42], coinciding well with the products of the reverse water-gas shift reaction. No obvious signals of gaseous $CH_4$ (3016 $cm^{-1}$) or other intermediate products were observed. The absence of peaks between 2800 and 2900 $cm^{-1}$ demonstrates the non-existence of formate species[28]. Millet et al. also reported that only a weak formate signal could be observed for small-sized Ni, which favored the production of CO, whereas, for aggregated Ni nanoparticles that can produce a significant amount of $CH_4$, formate seemed a key intermediate[40]. Therefore, the absence of formate and selective production of CO corresponds well with our small-sized Ni. The lack of formate species and bridged CO (1800−2000 $cm^{-1}$) impeded the further hydrogenation into $CH_4$[28]. This again demonstrates the high selectivity of CO for the $S_{Fe-Ni}$ catalyst. The reverse water-gas shift (RWGS) reaction commonly follows a direct dissociation route ($CO_2 \rightarrow *CO_2 \rightarrow *CO + *O$) or a formate route[28,43–45]. Since the signal of $*HCOO^-$ is negligible, the RWGS reaction on the $S_{Fe-Ni}$ catalyst is more likely to follow a dissociation route. The overall weak signals in the in situ DRIFTS spectra for $S_{Fe-Ni}$ are ascribed to its excellent absorption in the infrared region (Supplementary Fig. 13b), but the relatively poor signal-to-noise ratio might cause difficulty for interpretation. Therefore, additionally, we obtained in situ DRIFTS spectra for the $S_{SiO2-Ni}$ sample, which also has the characteristics of small-sized Ni but with more intense reflectance signals in the infrared light region (Supplementary Fig. 13b). It is a perfect substitute to verify the reaction route in this system. The in situ DRIFTS spectra of $S_{SiO2-Ni}$ show distinct signals of gaseous CO and still no other obvious intermediates (Supplementary Fig. 13c). This again suggests that the high possibility of a direct dissociation route for small-sized Ni in our catalysts as depicted above. Moreover, another control sample was prepared by

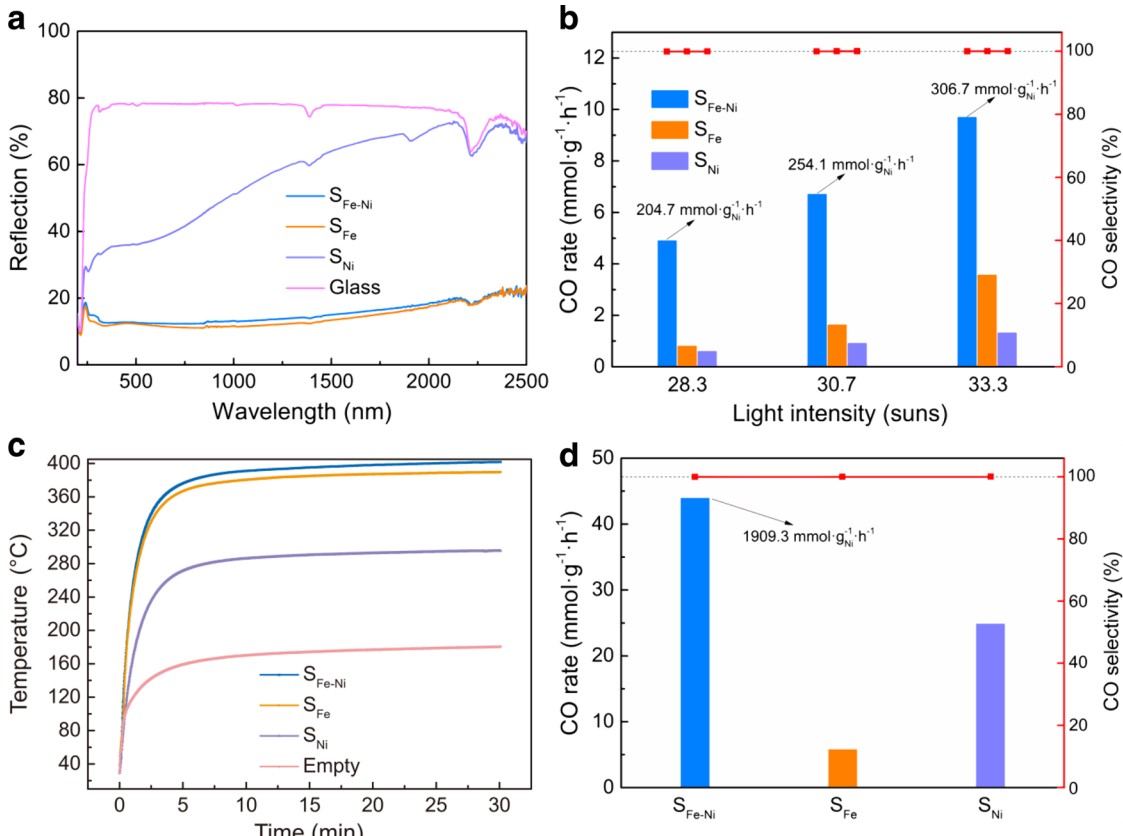

**Fig. 4 | Enhanced photothermal performance. a** Diffuse reflectance spectra of different catalysts. **b** CO production rates of different samples illuminated with a Xe arc lamp supplemented with a light concentrator in the flow reactor. The power of the lamp was set to be 190, 200, and 210 W, corresponding to the light intensities of 28.3, 30.7, and 33.3 suns, respectively. Three data points were collected to obtain the average CO rate after 10 min illumination for each sample. **c** Surface temperature profiles of different samples illuminated with concentrated light from a Xe arc lamp. The power of the lamp was set to be 210 W. The equilibrium temperatures under illumination for $S_{Fe-Ni}$, $S_{Fe}$, $S_{Ni}$, and without catalyst were 401.8, 389.6, 295.3, and 180.8 °C, respectively. **d** CO production rates of different samples in the batch reactor illuminated with a Xe arc lamp supplemented with a light concentrator. The light intensity was 28 suns. The CO rates with the unit of 'mmol·$g_{Ni}^{-1}$·$h^{-1}$' for $S_{Fe-Ni}$ in **b**, **d** were calculated based on the mass of Ni, and the activity of the isolated Fe component was deducted by subtracting the CO rate of $S_{Fe}$ under the same testing condition. The red data points in **b**, **d** correspond to the selectivity towards production of CO.

impregnating $Ni(NO_3)_2 \cdot 6H_2O$ on solid $SiO_2$ spheres ($Ni_{im}/SiO_2$) to show the difference between the small-sized Ni and Ni nanoparticles. The characteristic signals of crystal Ni in the XRD pattern (Supplementary Fig. 13d) confirm the nanoparticle state of Ni in $Ni_{im}/SiO_2$, which showed low selectivity towards CO (only 55.8%, Supplementary Table 2). The in situ DRIFTS spectra of $Ni_{im}/SiO_2$ show obviously characteristic peaks of gaseous $CH_4$ and linearly absorbed CO (2030 $cm^{-1}$)[46], which are not seen in the samples with small-sized Ni (Supplementary Fig. 13e). Therefore, the absence of intermediates verified from the in situ DRIFTS spectra of the samples with small-sized Ni can explain their much higher selectivity towards CO production, and accordingly insignificant production of $CH_4$. As to the basic kinetic studies, the results suggest that the CO production rate over the $S_{Fe-Ni}$ catalyst exhibits higher dependence on $CO_2$ (0.8) than $H_2$ (0.08), as shown by Supplementary Figs. 14 and 15, respectively. This indicates that the Ni surface is mostly covered by hydrogen instead of carbon species, which is in good agreement with the in situ DRIFTS results. When the pressure of $CO_2$ increases, the dissociation of $CO_2$ becomes easier, leading to promoted CO production.

**Photothermal catalytic performance**

It was found that $S_{Fe-Ni}$ and $S_{Fe}$ appeared to be black while the color of $S_{Ni}$ is pale (Supplementary Fig. 16). The diffuse reflectance spectra of these samples show that $S_{Fe-Ni}$ and $S_{Fe}$ have similar reflectance, which is less than 15%, spanning the ultraviolet, visible-light, and part of the near-infrared region (Fig. 4a). Even in the further end with the longer

wavelength, the reflectance of these two samples is not beyond 25%. In contrast, $S_{Ni}$ exhibits much higher reflectance throughout the full solar spectrum from 200 to 2500 nm. These results manifest that the Fe core plays an important role in improving the light-harvesting ability of $S_{Fe-Ni}$. The strong light absorption of $S_{Fe-Ni}$ should render it an excellent photothermal material, which allows the $CO_2$ hydrogenation reaction to be more sustainable with the aid of light[28,47–55].

The photothermal catalytic performances of the samples were first determined under illumination with a Xe arc lamp and without any heat supplied by the heating unit of the bulk reactor. Surprisingly, differently in the thermal test results, the yield of CO was only observed for $S_{Fe-Ni}$ while the other two samples had no catalytic activity (Supplementary Fig. 17). With the further assistance of a light concentrator, the production of CO could finally be detected for $S_{Fe}$ and $S_{Ni}$ (Fig. 4b). However, their CO yields were still very low, in contrast to the pronounced rates for $S_{Fe-Ni}$ under the same testing conditions. Typically, CO rates of 9.73, 3.59, and 1.35 mmol·$g_{cat}^{-1}$·$h^{-1}$ were obtained for $S_{Fe-Ni}$, $S_{Fe}$, and $S_{Ni}$, respectively, under concentrated 210-W illumination. Meanwhile, the production of $CH_4$ was still negligible in photothermal catalytic reactions as well (Supplementary Fig. 18), ensuring ultrahigh selectivity towards CO (>99.99%). These results indicate that both the Fe core and Ni sites are indispensable components in the nanoreactor for achieving considerable CO rates in photothermal catalytic $CO_2$ hydrogenation. Compared with $S_{Ni}$ that is devoid of Fe, the CO production rate of $S_{Fe-Ni}$ was significantly boosted, around 7 times, which can be ascribed to the excellent light-harvesting ability

and thereby high photo-to-thermal conversion efficiency of the Fe core. These are manifested by the fact that the equilibrium temperature of $S_{Fe-Ni}$ could be achieved at the high value of 402 °C under the illumination of concentrated light (Fig. 4c). The surface temperature of $S_{Fe}$ is close to that of $S_{Fe-Ni}$, which is far beyond 295 °C, the equilibrium temperature of $S_{Ni}$. This trend was also found under other illumination conditions (Supplementary Fig. 19). Therefore, the observed high rate is largely attributed to the heat provided by the Fe core under such light-only conditions. In comparison, $S_{Fe}$, which is with the Fe core but without Ni sites, could reach a similar high temperature under light but did not show a high reaction rate. This echoes with the thermocatalytic result that Ni was the major active site for the hydrogenation of $CO_2$. Notably, $S_{Fe-Ni}$ could achieve similar CO rates of ~5 mmol·g$^{-1}$·h$^{-1}$ in either the thermocatalytic process (500 °C) or the photothermal catalytic (190 W illumination assisted with a concentrator) process. This indicates the real local temperature of $S_{Fe-Ni}$ under this illumination was much higher than the apparent surface temperature of 361 °C measured by the thermocouple, further demonstrating the excellent photothermal conversion property of $S_{Fe-Ni}$. The increased local temperature and its catalytic boosting effect induced by $SiO_2$ encapsulation have also recently been discovered and discussed in detail by Cai et al.[56,57]. The heat insulation and infrared shielding effects of the $SiO_2$ sheath confine the photothermal energy of the Fe core within the catalyst, enabling a supra-photothermal effect. The thermocouple could only reach the surface of the outer-layer mesoporous silica, the temperature of which was lower than that inside the catalyst. The fact that $S_{Ni}$ only exhibited nearly 1/7 of the CO production rate of $S_{Fe-Ni}$ under the same light illumination condition (33.3 suns) indicates that photochemical contribution if there was any, was trivial. The significant CO rate of $S_{Fe-Ni}$ under light was still driven by the thermochemical reaction pathway, supported by the same product selectivity and similar activation energy as shown in thermal catalytic test results, shown in Supplementary Table 6 and Supplementary Fig. 20. Besides, the exponential dependence of the reaction rate on illumination intensity in Fig. 4b rather than a linear dependence, is a characteristic feature of thermally driven transformation[58]. The excellent photothermal catalytic performance of $S_{Fe-Ni}$ was further demonstrated by using commercial Cu-ZnO-Al$_2$O$_3$ as a control sample which shows a lower CO rate than $S_{Fe-Ni}$ under similar testing conditions (Supplementary Fig. 21). Another control experiment was also conducted by using only $CO_2$ and $N_2$ as the reactants, under which the production of CO is negligible in comparison to the $H_2$-existing atmosphere (Supplementary Fig. 22). This demonstrates that the production of CO originates from the reverse water-gas shift reaction rather than the Boudouard reaction ($C + CO_2 \rightarrow 2CO$).

Notably, the photothermal catalytic CO rate for $S_{Fe-Ni}$ could be further improved to be 44.1 mmol·g$^{-1}$·h$^{-1}$ by using a batch reactor, which is about 9 times the rate in the flow reactor under the same light intensity (Fig. 4b, d). At the same time, the selectivity was maintained to be nearly 100%. This improvement can be attributed to the sealed environment and the catalyst bed geometry that can reduce loss of heat (reduced thermoconvection compared with the flow reactor) and enhance light absorption (larger absorption area due to the better dispersion of catalyst on the glass fiber filter). The photothermal catalytic activity of $S_{Fe-Ni}$ was far beyond that of $S_{Fe}$ (5.9 mmol·g$^{-1}$·h$^{-1}$) in the batch reactor, again demonstrating that Ni is the major active component. This allows the CO production rate per weight of Ni to be calculated, which turned out to be as high as 1.9 mol·g$_{Ni}^{-1}$·h$^{-1}$. Meanwhile, the activity of the control sample $S_{Ni}$ in the batch reactor was also improved compared to the flow reactor result, but only around half of the activity of $S_{Fe-Ni}$, due to the lack of the central Fe heating core. The photothermal activity of our catalyst was also compared with those of various reported photothermal catalysts (Supplementary Table 7). The $S_{Fe-Ni}$ samples have higher or at least the same level of production rate compared with the listed catalysts. Overall, the

intentionally designed architecture of our nanoreactor catalyst and control tests have pinpointed the functions of the Fe core and the well-distributed Ni sites, and exhibited extraordinarily high catalytic selectivity at a high production rate.

## Cycle experiment

The concentration of Ni in the electroplating wastewater was 6576 mg L$^{-1}$ according to the inductively coupled plasma mass spectrometry (ICP-MS) results. Obviously, the Ni resources in the wastewater were affluent. In this work, the Ni wastewater could be utilized as an abundant source for preparing large amounts and multiple batches of $S_{Fe-Ni}$ catalysts. To confirm this concept, a batch of wastewater was treated by $S_{ad}$ repeatedly to determine how many times it could serve for the preparation of $S_{Fe-Ni}$. For each cycle of treatment, the fresh adsorbent was used, and the catalyst thus prepared was named as $S_{Fe-Ni}n$, in which n represents the cycle number. Supplementary Fig. 23 shows the concentration of the remaining Ni in the wastewater after numerous typical cycles (abbreviated as $C_{Ni}$-n; n represents the cycle number). After 15 cycles, $C_{Ni}$-15 was detected to be 224 mg L$^{-1}$, about 3% of the original value (6676 mg L$^{-1}$). Namely, 97% of the Ni resources in the wastewater were upcycled to prepare 15 batches of the active catalyst $S_{Fe-Ni}$. The loading amounts of Ni onto $S_{Fe-Ni}$ after these cycles were determined by ICP-MS (Supplementary Fig. 24). In the initial few cycles, the loading ratios of Ni for different $S_{Fe-Ni}$ samples were similar: ~2%. After the 9th cycle, the Ni loading began to decrease, which is ascribed to the lower maximum adsorption capacity under equilibrium when $C_{Ni}$-n drops. The loading ratio of Ni was 1% for $S_{Fe-Ni}15$, which is half of that for $S_{Fe-Ni}1$ (Supplementary Fig. 24 and Supplementary Table 4). The thermocatalytic property for three typical $S_{Fe-Ni}n$ (n = 3, 9, and 15) samples at 500 °C was evaluated (Supplementary Fig. 25). The CO production rates of $S_{Fe-Ni}3$ and $S_{Fe-Ni}9$ approached that of $S_{Fe-Ni}$, which should be attributed to their similar loading amount of Ni and catalyst structure. In contrast, $S_{Fe-Ni}15$ showed a much lower CO production rate (3.12 mmol g$_{cat}^{-1}$ h$^{-1}$), which was expected since its Ni loading was only half that of $S_{Fe-Ni}$. Nevertheless, this rate is still higher than that of $S_{Fe}$, again demonstrating the function of adsorbed Ni as the active sites. Besides, only a trace amount of $CH_4$ could be detected for all these samples, marking good repeatability in achieving high product selectivity via this cycling approach. Overall, 97% of Ni in the electroplating wastewater could be recycled for the preparation of $S_{Fe-Ni}$ catalysts through cycling adsorption and calcination, realizing the sustainable upcycling of waste Ni.

Despite the overall high upcycling efficiency through cycle adsorption, the drop of activity for the samples in the later cycles due to the lower loading of Ni might be a problem. We have clarified this in the revised manuscript, as the reviewer suggested. Moreover, we further propose a possible remedial strategy regarding this problem: performing adsorption in fresh wastewater again using the sample in the later cycles to achieve improved Ni loading. As a simple demonstration, we prepared a $S_{Fe-Ni}$ sample using diluted Ni wastewater (1 mL of wastewater diluted by 29 mL of milli-Q water) to simulate $S_{Fe-Ni}$ in the later cycles. Next, a second adsorption process in the fresh Ni wastewater was operated on this sample. The CO rate of the treated sample achieved was 5.3 mmol g$_{cat}^{-1}$ h$^{-1}$ which is similar to that of $S_{Fe-Ni}$ prepared using fresh Ni wastewater under the same testing condition, demonstrating the feasibility of the remedial strategy. Nevertheless, the second adsorption also made the upcycling procedure more time-consuming. Simpler and more cost-effective upcycling strategies should be developed in future works.

## $CO_2$ footprint and cost evaluation

Carbon neutrality is an urgent goal in the current age[59–65]. Many technologies for producing value-added products claim using $CO_2$ as the feedstock can reduce $CO_2$ emission, but this may not be necessarily true, since the power consumed to run the facility, to produce $H_2$, and

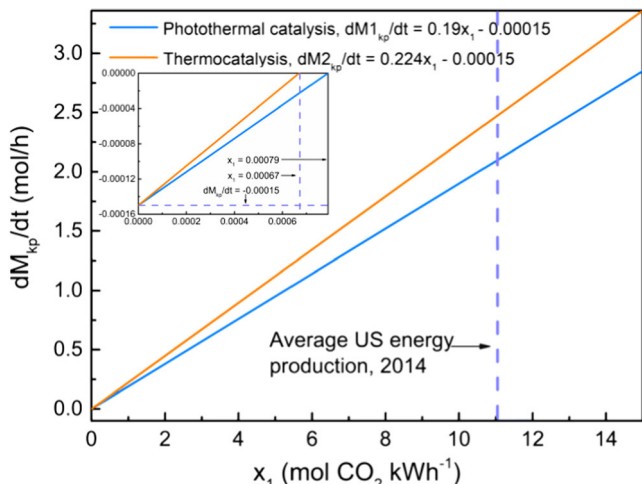

**Fig. 5 | Reduced CO₂ footprint in photothermal catalysis.** The dependence of net $CO_2$ reduction rate on $CO_2$ emission per kWh of electricity ($x_1$) for flow reactors based on the assumption that $H_2$ is $CO_2$-cost-free. The values of $a$, $c$, $m$, and $k_2T+b_2$ were set to be 0.19, 0.005, 0.03, and 0.224, respectively, according to the catalytic results and reactors used in this work. In the inset, the vertical dashed line shows the minimum $x_1$ which can achieve net $CO_2$ reduction while the horizontal dashed line represents the net $CO_2$ emission rates when the production of electricity is $CO_2$-free.

to drive the chemical reaction may generate an even larger $CO_2$ footprint than the one the technology aims to reduce[66]. Chen's group has recently demonstrated that using $CO_2$ to synthesize methanol either via thermocatalysis or electrocatalysis is still generating several mol of new $CO_2$ when one mol of $CO_2$ is consumed under the current US energy structure, far from reaching carbon neutrality[67]. Inspired by his quantitative study, we recently developed a universal evaluation approach to compare the net $CO_2$ emission rates between thermal and photothermal $CO_2$ hydrogenation using typical small-scale fix-bed reactors[68]. As shown in Fig. 5, net $CO_2$ emission rates (denoted as $dM/dt$) for $S_{Fe-Ni}$ under the thermocatalytic and photothermal conditions were calculated based on the catalytic results from this work and the criterions built in our previous work. To be specific, the value of $M$ equals the amount of emitted $CO_2$ after subtracting the consumed $CO_2$ in a catalytic process. The derivatives of $M$ in terms of t ($dM/dt$, representing net $CO_2$ emission rate) for flow reactors are shown as follows:

$$dM1_{kp}/dt = x_1a + nx_2 - mc \qquad (1)$$

$$dM2_{kp}/dt = (k_2T + b_2)x_1 + nx_2 - mc \qquad (2)$$

in which, $a$, $c$, $m$, $n$, $x_1$, $x_2$, and $k_2T+b_2$ represent the power consumption of the lamp (kW), the conversion rate of $CO_2$ (mol·g⁻¹·h⁻¹), the mass of the catalyst (g), the original feed amount of $H_2$ in a cycle (mol), $CO_2$ emission per kWh of electricity according to the US 2014 energy structure (mol), $CO_2$ produced per mol of $H_2$ (mol), and the electricity consumed to maintain the reactor at $T$ per hour (kWh), respectively. The subscript 'kp' denotes that the $H_2$ feed is calculated based on the kinetic parameters. Detailed calculation and value assignment are listed in the supplementary notes.

Owing to the high photothermal activity of $S_{Fe-Ni}$, the net $CO_2$ emission rates for the photothermal catalytic process (denoted as $dM1_{kp}/dt$) were obviously lower than that for the thermocatalytic process (denoted as $dM2_{kp}/dt$) when electricity was used to power the facilities (either the Xe arc lamp or the heating system), as shown in Fig. 5. For example, when considering the $CO_2$ emission per kWh of

electricity according to the reported US 2014 energy structure, the net $CO_2$ emission rates of using 30 mg of the $S_{Fe-Ni}$ catalyst was significantly reduced by 15% from 2.473 mol h⁻¹ (500 °C) to 2.097 mol h⁻¹ (190 W illumination supplemented with a light concentrator), while maintaining the same production rate of CO. Despite that these electricity-powered catalytic processes are still generating $CO_2$ rather than reducing the amount of $CO_2$, the values of $dM/dt$ for the photothermal catalytic process could be greatly reduced by increasing the usage of sustainable energy (sunlight, tide, wind, geothermal energy, etc.) to reduce $x_1$ (Fig. 5). When $x_1$ is 0.00079 mol $CO_2$ kWh⁻¹, the photothermal route can veritably achieve carbon neutrality, and at the same time produce fuel.

When it comes to practical utilization, the cost of Ni during catalyst preparation is also very important in addition to the $CO_2$ footprint. The common Ni resources used in the preparation of Ni catalysts include $Ni(NO_3)_2$·$6H_2O$, $NiCl_2$·$6H_2O$, and $NiSO_4$·$6H_2O$. The prices of these Ni resources range from 1.7 to 9.84 USD g⁻¹ (Supplementary Table 8). In contrast, the electroplating industries need to pay for the treatment of the Ni wastewater, contributing to a negative cost of Ni (−\$0.0019 g⁻¹ to −\$0.0028 g⁻¹) from it. This makes the wastewater from electroplating an ideal precursor for the preparation of the $S_{Fe-Ni}$ catalysts.

## Discussion

The essence of the work presented herein is inspired by the pre-science of Buckminster-Fuller, who in a 1971 Life Magazine interview said, "There is no energy crisis, just a crisis of ignorance." He imagined pollution as a feedstock to be harvested for its value rather than treated as the waste of a consumer society. This is the underpinning of our carefully designed adsorbent for the Ni tailings in electroplating wastewater. They can be given new life in a recycle-upcycle process that takes the Ni from a toxic grave to a sustainable cradle, a kind of inverse life cycle paradigm to create a cost-effective eco-friendly photothermal $CO_2$ catalysis process. We envision many 'pollution solutions' that could benefit from the recycle-upcycle concepts and principles of the genre delineated herein, turning plastic, paper, mining, food, and electronic waste into sustainable consumer products.

## Methods
### Materials and chemicals
All the chemicals were used as received without further purification. Commercial Cu-ZnO-Al₂O₃ was purchased from Sichuan Shutai Chemical Technology Co., Ltd. Tetraethyl orthosilicate (TEOS, > 96%), N,N-diisopropylethylamine (>99%), and hexadecyl trimethyl ammonium bromide (CTAB, >98%) were purchased from TCI. Ferric chloride hexahydrate (FeCl₃•6H₂O, reagent grade), nickel nitrate hexahydrate (Ni(NO₃)₂•6H₂O, ≥97%), N¹-(3-trimethoxysilylpropyl) diethylenetriamine (≥98%), and PSSMA [poly(4-styrenesulfonic acid-co-maleic acid) sodium salt] were purchased from Sigma–Aldrich. Sodium acetate anhydrous (99%), hydrochloric acid (36%–38%, analytical reagent), ethylene glycol (99.5%), ethanol (GR, ≥99.8%), and ammonium hydroxide solution (NH₃•H₂O, 28 wt %) were purchased from Energy Chemical, Enox, J&K scientific, Sinopharm Chemical Reagent Co., Ltd, and Macklin, respectively. Milli-Q water (Millipore, 18.2 MΩ cm at 25 °C) was used in all experiments.

### Synthesis of Fe₃O₄ colloidal nanocrystal clusters (CNC)
Fe₃O₄ CNCs were synthesized according to a reported recipe[69,70]. For a typical batch, 7.5 g of PSSMA was dissolved in 300 mL of ethylene glycol with magnetic stirring. 8.1 g of FeCl₃•6H₂O and 22.5 g of sodium acetate were then added to the mixture under continuous stirring until the solution turned homogeneously red-brown. The mixture was then sealed in a Teflon-lined stainless-steel autoclave and heated at 200 °C for 10 h. When cooled to room temperature, the dark precipitates were

isolated from the solution by a magnet and washed with Milli-Q water and ethanol alternately six times, and finally dispersed in ethanol to form a suspension (concentration: 10 mg mL$^{-1}$).

### Synthesis of CNC@SiO$_2$@mSiO$_2$-NH$_2$

The CNC nanoparticles were coated with a thin layer of dense silica via a modified Stöber method[71,72]. Briefly, 20 mL of the CNC suspension was diluted by ethanol (60 mL) and Milli-Q water (12 mL). The mixture was then sonicated for 30 min, followed by the addition of ammonium hydroxide solution (10 mL), and TEOS (500 μL) sequentially. The reaction vessel was then stirred in a shaking bed (400 rpm, 30 °C) for 1 h. The obtained nanoparticles (denoted as CNC@SiO$_2$) were washed twice with ethanol under centrifugation and redispersed in Milli-Q water to form a suspension (concentration: 40 mg mL$^{-1}$). 2.5 mL of the CNC@SiO$_2$ suspension was added to 20 ml of the prepared CTAB solution (0.9 g of CTAB dissolved in a mixture of 100 mL of ethanol and 300 mL of Milli-Q water). The suspension was sonicated and stirred, each for 20 min, followed by the sequential addition of ammonium hydroxide solution (125 μL), and TEOS (175 μL). The mixture was stirred for another 4 h in a shaking bed (400 rpm, 30 °C). The obtained nanoparticles were then separated and redispersed in acetone and refluxed at 80 °C for 48 h. The refluxing procedure was repeated 3 times. The obtained products (denoted as CNC@SiO$_2$@mSiO$_2$) were then washed with ethanol several times and dried in a vacuum oven. One hundred milligrams of the CNC@SiO$_2$@mSiO$_2$ powder was dispersed in ethanol (40 mL). 0.5 ml of N$^1$-(3-trimethoxysilylpropyl) diethylenetriamine and 0.1 ml of N,N-diisopropylethylamine were then added to the suspension. The mixture was stirred in a shaking bed (400 rpm, 30 °C) for 12 h and then washed with Milli-Q water and ethanol six times alternately. The obtained nanoparticles (denoted as CNC@SiO$_2$@mSiO$_2$-NH$_2$, abbreviated as S$_{ad}$) were dried under vacuum.

### Synthesis of SiO$_2$@mSiO$_2$-NH$_2$

Silica nanospheres with a similar size of CNC@SiO$_2$ were prepared via a modified Stöber method[71,72]. 3.6 mL of Milli-Q water, 24 mL of ethanol, and 800 μL of ammonium hydroxide solution were mixed, followed by the addition of 265 μL of TEOS. The mixture was stirred in a shaking bed (400 rpm, 30 °C) for 1 h. The resulting nanoparticles were washed twice with ethanol and were finally dispersed in Milli-Q water to form a suspension (concentration: 26 mg mL$^{-1}$). 2.5 mL of the SiO$_2$ suspension was added to 20 ml of the prepared CTAB solution (0.9 g of CTAB dissolved in a mixture of 100 mL of ethanol and 300 mL of Milli-Q water). The suspension was sonicated and stirred, each for 20 min, followed by the sequential addition of ammonium hydroxide solution (125 μL), and TEOS (175 μL). The mixture was stirred for another 4 h in a shaking bed (400 rpm, 30 °C). The obtained nanoparticles were then separated and redispersed in acetone and refluxed at 80 °C for 48 h. The refluxing procedure was repeated three times. The obtained products (denoted as SiO$_2$@mSiO$_2$) were then washed with ethanol several times and dried in a vacuum oven. 100 mg of the SiO$_2$@mSiO$_2$ powder was dispersed in ethanol (40 mL), followed by the addition of N$^1$-(3-trimethoxysilylpropyl) diethylenetriamine (0.5 ml) and N,N-diisopropylethylamine (0.1 ml). The mixture was stirred in a shaking bed (400 rpm, 30 °C) for 12 h and then washed with Milli-Q water and ethanol 6 times alternately. The obtained nanoparticles (denoted as SiO$_2$@mSiO$_2$-NH$_2$) were dried under vacuum.

### Synthesis of amino-grafted solid SiO$_2$ (S$_{SiO2-NH2}$)

Solid silica nanospheres (S$_{SiO2}$) were prepared via a modified Stöber method[71,72]. Milli-Q water (3.6 mL), ethanol (24 mL), and ammonium hydroxide solution (3 mL) were mixed, followed by the addition of 150 μL of TEOS. The mixture was stirred in a shaking bed (400 rpm, 30 °C) for 1 h. The resulting nanoparticles were washed twice with ethanol and were finally dried under vacuum. 100 mg of the S$_{SiO2}$ powder was dispersed in ethanol (40 mL), followed by addition of N$^1$-(3-trimethoxysilylpropyl) diethylenetriamine (0.5 ml) and N,N-diisopropylethylamine (0.1 ml). The mixture was stirred in a shaking bed (400 rpm, 30 °C) for 12 h and then washed with Milli-Q water and ethanol six times alternately. The obtained nanoparticles (S$_{SiO2-NH2}$) were dried under vacuum.

### Preparation of the catalysts

Two hundred milligrams of the dried S$_{ad}$ or SiO$_2$@mSiO$_2$-NH$_2$ or S$_{SiO2-NH2}$ powder was mixed with 10 mL of the electroplating wastewater, the compositions of which are listed in Supplementary Table 3. The suspension was sonicated and then transferred to a shaking bed (400 rpm, 30 °C), subjected to stirring overnight. The adsorbents together with the adsorbed metal ions were collected via magnetic separation and then dried in a vacuum oven. The dried samples were calcined at 500 °C for 1 h in a muffle furnace to remove the organic pollutants, followed by reduction at 600 °C for 2 h in H$_2$ atmosphere. The reduction temperature, under which the metal precursors could be fully reduced, was determined by the TPR results (Supplementary Fig. 26). The final products were denoted as S$_{Fe-Ni}$, S$_{Ni}$, and S$_{SiO2-Ni}$, corresponding to the S$_{ad}$, SiO$_2$@mSiO$_2$-NH$_2$ and the S$_{SiO2-NH2}$ support, respectively. The loading amounts of Ni and Fe for different samples are listed in Supplementary Table 4. A control sample was prepared by replacing the electroplating wastewater with manmade Ni$^{2+}$ solution from reagent Ni(NO$_3$)$_2$·6H$_2$O. The compositions of the prepared catalysts are listed in Supplementary Tables 4 and 5.

In a cycle adsorption process, 10 mL of wastewater was mixed with S$_{ad}$. After adsorption equilibrium, the spent S$_{ad}$ was separated from the solution for the following calcination process, and the same amounts of new S$_{ad}$ were put into the treated wastewater. The procedure was repeated multiple times.

### Thermocatalytic tests

Thermocatalytic CO$_2$ hydrogenation under atmospheric pressure was performed in a quartz tube flow reactor (CEL-GPPCM, BEIJING CHINA EDUCATION AU-LIGHT CO., LTD., Supplementary Fig. 27) at a certain temperature (300, 350, 400, 450, or 500 °C). Prior to the reaction, the powder catalysts (<80 mesh) were reduced under H$_2$ (20 mL min$^{-1}$) at 600 °C for 2 h. The flow rates of CO$_2$, H$_2$, and N$_2$ were set at ~2.5, 2.5, and 5 mL min$^{-1}$, respectively. The products at the reactor outlet were detected with an online gas chromatographer (Agilent 8860) equipped with a thermal conductivity detector (TCD) and a flame ionization detector (FID).

In-situ DRIFTS experiments were performed on a Bruker VERTEX 70 v Spectrometer with a mercury cadmium telluride detector cooled with liquid nitrogen. Approximately 20 mg of as-reduced S$_{Fe-Ni}$, S$_{SiO2-Ni}$, or Ni$_{im}$/SiO$_2$ catalyst (<80 mesh) was packed into an in situ cell (BEIJING OPERANDO TECHNOLOGY CO., LTD.). Prior to the reaction, the catalyst was reduced in a H$_2$/N$_2$ mixture gas (2.5/17.5 mL/min) at 500 °C for 30 min. The inlet flow in the cell was then switched to N$_2$ (17.5 mL/min) for 30 min, and the temperature was kept at 500 °C. After obtaining the background spectra, the inlet flow was switched to a CO$_2$/H$_2$ mixture gas (2.5/2.5 mL/min). The adsorption species on the surface of catalysts were detected online.

Rate-order experiments were carried out with 30 mg of the S$_{Fe-Ni}$ catalyst (<80 mesh) at 500 °C under a constant flow rate (40 mL/min). The internal and external diffusions were eliminated based on a reported recipe (see the details in the supplementary notes, Supplementary Figs. 28 and 29). To study the rate order for H$_2$, the concentration of feed CO$_2$ was kept at 12.5%, while the concentration of feed H$_2$ gradually decreased from 67.5% to 42.5% and then gradually increased back to 67.5%, and N$_2$ was used as the balance component. The CO rates in the downslope and uphill stages under the same conditions were averaged for the calculation of rate order for H$_2$. Similarly, to study the rate order for CO$_2$, the concentration of feed H$_2$

was kept at 75%, while the concentration of feed $CO_2$ was gradually increased from 7.5 to 11.25% and then gradually decreased to 7.5%, and $N_2$ was used as the balance component. The CO rates in the upslope and downslope stages under the same conditions were averaged for the calculation of the rate order for $CO_2$. In all rate-order experiments, the conversions for $H_2$ and $CO_2$ were always kept below 5%.

### Photothermal catalytic tests in the flow reactor
The photothermal tests were conducted similarly to the thermo-catalytic tests, except that a 300-W Xe arc lamp (PF300-T8E, BEIJING CHINA EDUCATION AU-LIGHT CO., LTD.) rather than heating mod-ulate was used as the energy source. The light intensity was measured using an optical power meter (PL-MW2000, Beijing Perfectlight Technology Co., Ltd.). The relationship between the power of the lamp and light intensity is listed in Supplementary Fig. 30

### Photothermal catalytic tests in the batch reactor
The gas-phase photothermal catalytic experiments were conducted in a batch reactor (CEL-HPR100T+) with an inner volume of 100 mL. A group of glass slides was put inside the reactor to support the catalyst. The final effective volume of the reactor was 57.77 mL. A 300 W Xe arc lamp was used to illuminate the catalysts with the assistance of a concentrator. For all the catalysts, Samples (9 mg) were dispersed in ethanol and then transferred into a glass fiber filter through drop-casting. The catalyst film was then dried under a vacuum. After the loading of the catalyst film, the reactor was degassed first. A mixture of $CO_2$ and $H_2$ (1:1) was used to purge the reactor three times. The reactor was sealed when the pressure reached 1 bar. The lamp was then turned on to initiate the photo-thermal catalytic reaction. After 10 min illumination, product gases were analyzed with a gas chromatographer (Agilent 8860) equipped with a thermal conductivity detector (TCD) and a flame ionization detector (FID).

### Characterization
STEM images and EDX mappings were obtained in a double aberration-corrected transmission electron microscope (FEI Titan Themis G2) operated at 300 kV with a HAADF detector (collection angle range of 48–200 mrad) and Super-X EDX detector. Transmission electron microscopy (TEM) images were obtained with an FEI-Tecnai F20 (200 kV) transmission electron microscope. The XAS experiments at the Ni K-edge were performed at the Shanghai Synchrotron Radiation Facility (SSRF,11B), and the intensities of the two $S_{Fe-Ni}$ samples were multiplied by five for the EXAFS spectra. The metal content of different samples was measured by an Inductively coupled plasma optical emission spectrometer (ICP-OES) (i CAP Pro X, Thermofisher) or an Inductively coupled plasma source mass spectrometer (ICP-MS) (Aurora M90, Jenoptik). The temperature-programmed reduction (TPR) results were recorded with an infrared spectrometer (Thermo-Fisher Nicolet iS 50) equipped with a Deuterated Triglycine Sulfate (DTGS) detector.

### Reporting summary
Further information on research design is available in the Nature Research Reporting Summary linked to this article.

## Data availability
The datasets generated during and/or analyzed during the current study are available from the corresponding author on reasonable request. Source data are provided with this paper.

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

## Acknowledgements

W.S. thanks the support from the National Key R&D Program of China (2021YFF0502000), National Natural Science Foundation of China (51902287), the U of T-ZJU Joint Seed Fund, the Fundamental Research Funds for the Central Universities (226-2022-00159, 226-2022-00200), Jiangsu Key Laboratory for Carbon Based Functional Materials & Devices, Soochow University, and State Key Laboratory of New Textile Materials and Advanced Processing Technologies (FZ2020020). D.Y. acknowledges the financial support from National Natural Science Foundation of China (61721005). L.H. thanks the support by National Natural Science Foundation of China (52172221 and 51920105005), the Natural Science Foundation of Jiangsu Province (BK20200101), 111 Project, and the Collaborative Innovation Centre of Suzhou Nano Science & Technology. G.A.O. is the Government of Canada Tier 1 Research Chair in Materials Chemistry and Nanochemistry, and he acknowledges the financial support provided by the Natural Sciences and Engineering Research Council of Canada (NSERC). The authors thank the support from SSRF (11B) for the XAS experiments.

## Author contributions

W.S., S.W., D.Z., L.H., D.Y., and G.A.O. conceived and designed the experiments. S.W. and D.Z. carried out the synthesis and catalysis experiments. W.W. and J.H. carried out the HAADF-STEM experiments. J.Z. carried out and analyzed the XAS experiments. K.F. analyzed the XAS results. Z.W. carried out the TEM characterization. B.D. carried out the ICP experiments. Z.L. carried out part of the thermocatalytic experiments. W.S., G.A.O., and S.W. wrote the paper. All authors commented on the final manuscript.

## Competing interests

The authors declare no competing interests.
