## [Peer Review File · Nature Communications]

Grave-to-Cradle Upcycling of Ni from Electroplating
Wastewater to Photothermal CO₂ CatalysisREVIEWER COMMENTS

Reviewer #1 (Remarks to the Author):

The manuscript shows a way of for upcycling waste Ni from electroplating wastewater into a photothermal catalyst for CO₂ hydrogenation to CO. The topic and results are interesting and are likely to attract the interest of the journal readers. However, I am not fully convinced by the usefulness of this approach to make high performance catalysts. I think the paper can be considered for publication if following suggestions/comments are addressed.

1) Standard catalysts including Cu-based and Fe-Cr-based industrial catalysts as well as reported Ni-based catalysts for hydrogenation of CO₂ to CO should be tested to show the developed catalyst is truly effective for this process. For Ni-based CO-producing catalysts, please at least check the following papers.

Journal of Rare Earths, 2013, 31, 559, iScience, 2019, 17, 315

2) Related to above, catalytic activities obtained should be compared with reported values.

3) Description of reaction mechanisms including information on catalytically active sites should be improved. In situ/operando spectroscopic measurements and kinetic studies should be performed.

Reviewer #2 (Remarks to the Author):

This manuscript reports the use of core-shell magnetic nanoparticles having Fe magnetite as core and a shell of porous silica modified with aminopropyl silyl groups to capture Ni ions as photocatalyst for photothermal reverse water gas shift with a high CO rate. Incidentally, the Ni of the photocatalyst can be obtained from electroplating waste water, but this could rise some issues that have not been considered in the manuscript. Publication in Nature Commun. is recommended based on the current interest in light-assisted CO₂ reactions, the novelty of the core-shell structuration and the photocatalyst performance. However, the authors have to consider the following comments:

- As commented in the introduction, electroplating waste water is a complex mixture containing many different components. Conceptually, this complexity is detrimental for a reliable catalyst preparation in which minute variations may influence considerably the activity due to the presence of poisons or competing species. What can be the influence of Cu, Cr and other metals present in the electroplating waste waters. A critical comment on the concept of the paper is needed.
- Rather than qualitative statements on the role of the middle mantle protecting magnetite from acids, the authors should present experimental data of stability of the iron core against corrosion of acids at various concentrations over the time. It is unclear, why the core can be reduced by H₂ and are protected from protons.
- I do not understand the nomenclature of SFe-Ni. It confuses the chemical reader with the presence of sulfur in the material. In addition, the chemical composition of the catalyst, particularly the Ni content and the presence of other metal impurities should be given.
- The previous point is in connection of how to present the results. I think that giving the CO production as mol/gNi can be misleading. Data should be given, at least, per total mass of the photocatalyst in abstract and other parts of the manuscript.
- Specifically in page 10, line 231 a CO productivity rate of 5 mmol/gcat h for a reaction carried out at 500 oC is not an impressive number and it is in fact below many other thermal catalysts at this high temperature. Even more, the value of 4.87 mmol/gcat h indicated in page 11, line 264 is also low compared to the state of the art. In this part, the surface illuminated by the beam of 190 W must be given to have an idea of the number of suns used in this measurement.
- In page 10, lines 226-228, the reader is disoriented about whether or not the experiments have been carried out with a photocatalyst prepared using real electroplating waste water. If this were the case, then, to have reproducible results, full analytical data of the material must be given.
- I do not agree that bed configuration is the reason why the flow reaction gives higher CO production, The authors should consider the influence of H₂O in the batch reaction, poisoning the catalyst, while in

flow, the H₂O concentration could be steady during the process. Since the negative influence of H₂O on the photothermal reaction has been reported, the authors have to perform measurements in the presence of controlled amounts of H₂O in the feed and determine the influence of the presence of this product.

- The section of cycle experiments is less convincing, since the results could vary depending on the real source. There are currently various electroplating systems with various compositions. However, from the catalysis point of view, the worrying point is that over time, the composition of the catalyst will vary due to the different percentages of adsorbed metals. Here in the text, only Ni adsorption is commented, but the other metal components will also change in proportion upon cycle. I am not thinking that the concept could not work, I am only saying that the situation is far more complex than presented in this simple case.

In my opinion, this paper is publishable in Nature Commun after revision considering the current interest on photothermal CO₂ conversion, the structure of the photocatalyst and unexpected product selectivity. I also valued the section on CO₂ footprint of the process. However, the authors should take into account the above points that should improve the completeness of the study.

Reviewer #3 (Remarks to the Author):

In this article, real waste Ni from the electroplating industry was demonstrated as an effective resource for the advanced technology of photothermal CO₂ catalysis. The authors bridged the magnetic adsorption technique with photothermal catalyst design, and constructed a planet earth-mimic structure which indeed has the "greenhouse effect" that is beneficial for photothermal CO₂ catalysis. This is a smart study and it targets multiple hot issues which will draw broad interest, so I recommend publication after addressing my queries:

1. The authors have employed mesoporous SiO₂. What is the rationale besides the large surface area? Is the small size of Ni attributed to the confinement of the mesopores?
2. The authors mentioned that more Ni could be seen near the outer part but no serious aggregation was found, after the long-term test. Although the TEM image shows some evidence, XRD result after the test should be provided to prove that the Ni was still highly dispersed without forming nanoparticles.
3. Why was Ni slightly oxidized after the photothermal test?
4. Can this upcycling approach be extended to treating other kinds of wastewater, e.g. containing Cu cations?
5. Nickel has showed a wide range of applications in photothermal catalysis. The authors can add some references of nickel in introduction, such as Nat. Commun. 12, 3996, (2021), Nat. Commun. 10, 2359, (2019).

Response to Reviewer #1:

General Comments: The manuscript shows a way of for upcycling waste Ni from electroplating wastewater into a photothermal catalyst for CO₂ hydrogenation to CO. The topic and results are interesting and are likely to attract the interest of the journal readers. However, I am not fully convinced by the usefulness of this approach to make high performance catalysts. I think the paper can be considered for publication if following suggestions/comments are addressed.

Q1. Standard catalysts including Cu-based and Fe-Cr-based industrial catalysts as well as reported Ni-based catalysts for hydrogenation of CO₂ to CO should be tested to show the developed catalyst is truly effective for this process. For Ni-based CO-producing catalysts, please at least check the following papers. Journal of Rare Earths, 2013, 31, 559, iScience, 2019, 17, 315

Author Reply: We thank the reviewer for raising this important point. In this work, we aim to construct a high-performance photothermal catalyst rather than a thermal catalyst. As instructed by the reviewer to compare with industrial catalysts, the industrial Cu-ZnO-Al₂O₃ catalyst was used as a control sample to demonstrate the excellent photothermal catalytic performance of our designed catalyst (**Fig. S21**). The S_{Fe-Ni} sample showed higher CO rate than Cu-ZnO-Al₂O₃ under similar testing conditions. Moreover, the photothermal activity of our catalyst was also compared with those of various reported photothermal catalysts (**Table S7**). The S_{Fe-Ni} samples have higher or at least the same level of production rate compared with the listed catalysts. This sentence has been added to the third paragraph in the ‘Photothermal catalytic performance’ part.

A sentence has been added to the second paragraph in the ‘Photothermal catalytic performance’ part as follows: The excellent photothermal catalytic performance of S_{Fe-Ni} was further demonstrated by using commercial Cu-ZnO-Al₂O₃ as a control sample which shows a lower CO rate than S_{Fe-Ni} under the same testing condition (**Fig. S21**).

A figure has been added to the supporting information as **Fig. S21**. The orders of other figures have been updated.

A table has been added to the supporting information as **Table S7**. The orders of other tables have been updated.

The listed two references are good examples of Ni based catalysts for CO₂ reduction, which has been added to the introduction part as ref [29] and ref [30]. The orders of other references have been updated.

Figure S21. Photothermal catalytic CO production rates of Cu-ZnO-Al₂O₃ (testing condition: 1.5 mg of commercial Cu-ZnO-Al₂O₃ was diluted by 28.5 mg of commercial SiO₂, CO₂/H₂/N₂ = 2.5/2.5/5 mL/min, ambient pressure) and S_{Fe-Ni} (testing condition: 30 mg, CO₂/H₂/N₂ = 2.5/2.5/5 mL/min, ambient pressure). The Cu weight ratio of commercial Cu-ZnO-Al₂O₃ was determined by ICP-OES to be 42.9%. The diluted Cu-ZnO-Al₂O₃ sample (2.1 wt% Cu) in the testing condition has similar weight ratio of active metal as S_{Fe-Ni} (2.0 wt% Ni). The samples were illuminated with a Xe arc lamp supplemented with a light concentrator in the flow reactor. The light intensity was set to be 26.1 suns. The CO rates with the unit of 'mmol·g_{metal}⁻¹·h⁻¹' for S_{Fe-Ni} was calculated based on the mass of Ni, and the activity of the isolated Fe component was deducted by subtracting the CO rate of S_{Fe} under the same testing condition.

Table S7. Comparison of the catalytic performance of our catalyst and the other reported photothermal catalysts tested in batch reactors.

Catalyst	Metal [wt%]	CO ₂ : H ₂	Light source	Light intensity (mW/cm ²)	CO selec.	R _{max} (mol g _{metal} ⁻¹ h ⁻¹)
Ni/Al ₂ O ₃ ⁷	2.1	1:4	300 W Xe light (UV-Vis-NIR)	N/A	0.95%	2.3
Co/Al ₂ O ₃ ⁷	2.5	1:4	300 W Xe light (UV-Vis-NIR)	N/A	0.49%	0.9
Pd/Al ₂ O ₃ ⁷	2	1:4.1	300 W Xe light (UV-Vis-NIR)	N/A	1.36%	0.53
Pt/Al ₂ O ₃ ⁷	2.4	1:4.1	300 W Xe light (UV-Vis-NIR)	N/A	84.85%	0.47
Ir/Al ₂ O ₃ ⁷	2.8	1:4.1	300 W Xe light (UV-Vis-NIR)	N/A	36.74%	0.05
Ni/N _{5,0} -CeO ₂ ⁸	10	1:1	300 W Xe light (UV-Vis-NIR)	2110	~100%	0.209
Ru/TiNT ⁹	11.2	1:4	Newport solar simulator	150 ± 210°C	N/A	0.1107
Co@CoN&C-1 ¹⁰	76.3	1:1	300 W Xe light	N/A	91.1%	0.17
Ni/80Ce-20Ti ₁ SG ¹¹	10	1:4	300 W Xe light	N/A	N/A	0.17
Pd@Nb ₂ O ₅ ¹²	0.5	1:1	300 W Xe light	2500	N/A	0.98
S _{Fe-Ni} (this work)	Ni: 2	1:1	300 W Xe light (UV-Vis-NIR)	2800	> 99.8%	1.9 [*]

*The CO rates were calculated based on the mass of Ni, and the activity of the isolated Fe component was deducted by subtracting the CO rate of S_{Fe} under the same testing condition.

Q2. Related to above, catalytic activities obtained should be compared with reported values.

Author Reply: As depicted in Q1, the photothermal catalytic activities of our S_{Fe-Ni} sample had been compared with reported values (Table S7).

Q3. Description of reaction mechanisms including information on catalytically active sites should be improved. In situ/operando spectroscopic measurements and kinetic studies should be performed.

Author Reply: We thank the reviewer for these valuable suggestions. As instructed by the reviewer, the catalytic pathway was studied through the in-situ diffuse reflectance infrared Fourier transform spectroscopy (DRIFTS) experiments and the kinetic studies (Fig. S13-S15). The in-situ DRIFTS spectra of S_{Fe-Ni} (Fig.S13a) show weak signals of gaseous CO (2111 and 2174 cm⁻¹)^{28,38}, coinciding well with the products of the reverse water-gas shift reaction. No obvious signals of gaseous CH₄ (3016 cm⁻¹) or other intermediate products were observed. The absence of peaks between 2800-2900 cm⁻¹ demonstrate the non-existence of formate species.²⁸ Millet *et al* also reported that only weak formate signal could be observed for small-sized Ni which favored production of CO, whereas for aggregated Ni nanoparticles which can produce significant amount of CH₄, formate seemed an key intermediate.³⁶ Therefore, the absence of formate and selective production of CO correspond well with our small-sized Ni. The lack of formate species and bridged CO (1800-2000 cm⁻¹) impeded the further hydrogenation into CH₄.²⁸ This again demonstrates the high selectivity of CO for the S_{Fe-Ni} catalyst. The reverse water-gas shift (RWGS) reaction commonly follows a direct dissociation route (CO₂ → *CO₂ → *CO + *O) or a formate route.^{28,39}

⁴¹ Since the signal of $^*HCOO^-$ is negligible, the RWGS reaction on the S_{Fe-Ni} catalyst are more likely to follow a dissociation route. The overall weak signals in the in-situ DRIFTS spectra for S_{Fe-Ni} are ascribed to its excellent absorption in the infrared region (**Fig. S13b**), but the relatively poor signal to noise ratio might cause difficulty for interpretation. Therefore, additionally we obtained in-situ DRIFTS spectra for the S_{SiO_2-Ni} sample which also has the characteristics of small-sized Ni but with more intense reflectance signals in the infrared light region (**Fig. S13b**). It is a perfect substitute to verify the reaction route in this system. The in-situ DRIFTS spectra of S_{SiO_2-Ni} show distinct signals of gaseous CO and still no any other obvious intermediates (**Fig. S13c**). This again suggests that the high possibility of a direct dissociation route for small-sized Ni in our catalysts as depicted above. Moreover, another control sample was prepared by impregnating $Ni(NO_3)_2 \cdot 6H_2O$ on solid SiO_2 spheres (Ni_{im}/SiO_2) to show the difference between the small-sized Ni and Ni nanoparticles. The characteristic signals of crystal Ni in the XRD pattern (**Fig. S13d**) confirm the nanoparticle state of Ni in Ni_{im}/SiO_2 which showed low selectivity towards CO (only 55.8%, **Table S2**). The in-situ DRIFTS spectra of Ni_{im}/SiO_2 show obviously characteristic peaks of gaseous CH_4 and linearly absorbed CO (2030 cm^{-1})⁴² which are not seen with the samples with small-sized Ni. Therefore, the absence of intermediates verified from the in-situ DRIFTS spectra of the samples with small-sized Ni can explain their much higher selectivity towards CO production, and accordingly insignificant production of CH_4 . As to the basic kinetic studies, the results suggest that the CO production rate over the S_{Fe-Ni} catalyst exhibits higher dependence on CO_2 (0.8) than H_2 (0.08), shown by **Fig. S14** and **Fig. S15**, respectively. This indicates that the Ni surface is mostly covered by hydrogen instead of carbon species, which is in good agreement with the in-situ DRIFTS results. When the pressure of CO_2 increases, the dissociation of CO_2 becomes

easier, leading to promoted CO production. This paragraph has been added to the ‘Thermocatalytic performance’ part as the fifth paragraph.

We conducted the in-situ DRIFTS experiment multiple times. No obvious intermediate products besides the gaseous CO could be observed in all of these tests for the samples with small-sized Ni.

Three figures have been added to the supporting information as **Fig. S13-15**. The orders of other figures have been updated.

The details of the in-situ DRIFTS experiment and the rate-order experiments have been added to the ‘Thermocatalytic tests’ part.

The details of the elimination of internal and external diffusion in the rate order experiments have been added to the supplementary notes, **Fig. S28 and Fig. S29**.

Figure S13. (a) In-situ DRIFTS spectra of S_{Fe-Ni} . **(b)** Diffuse reflectance spectra of S_{Fe-Ni} and S_{SiO_2-Ni} from 600-4000 cm^{-1} . KBr was used as the referenced background. **(c)** In-situ DRIFTS spectra of S_{SiO_2-Ni} . **(d)** XRD pattern of solid SiO_2 supported Ni nanoparticles (Ni_{im}/SiO_2) and the standard Ni (JCPDS 04-0850). **(e)** In-situ DRIFTS spectra of Ni_{im}/SiO_2 . Test conditions of all samples in the in-situ DRIFTS experiments: ~ 20 mg of catalyst, $CO_2/H_2 = 2.5/2.5$ mL/min, ambient pressure.

Figure S14. CO₂ rate order for S_{Fe-Ni}. CO₂ concentration was varied between 7.5% and 11.25% while H₂ concentration remained at 75% with conversions below 5%.

Figure S15. H₂ rate order for S_{Fe-Ni}. H₂ concentration was varied between 42.5% and 67.5% while H₂ concentration remained at 12.5% with conversions below 5%.

Response to Reviewer #2:

General Comments: This manuscript reports the use of core-shell magnetic nanoparticles having Fe magnetite as core and a shell of porous silica modified with aminopropyl silyl groups to capture Ni ions as photocatalyst for photothermal reverse water gas shift with a high CO rate. Incidentally, the Ni of the photocatalyst can be obtained from electroplating waste water, but this could rise some issues that have not been considered in the manuscript. Publication in Nature Commun. is recommended based on the current interest in light-assisted CO₂ reactions, the novelty of the core-shell structuration and the photocatalyst performance. However, the authors have to consider the following comments:

Q1. As commented in the introduction, electroplating waste water is a complex mixture containing many different components. Conceptually, this complexity is detrimental for a reliable catalyst preparation in which minute variations may influence considerably the activity due to the presence of poisons or competing species. What can be the influence of Cu, Cr and other metals present in the electroplating waste waters. A critical comment on the concept of the paper is needed.

Author Reply: As **Table S3** shows, the electroplating wastewater used in our work contain many other ions besides Ni. Nevertheless, the concentrations of these ions are relatively low in comparison to Ni²⁺. To verify the influence of these ions as instructed by the reviewer, a synthetic solution made from reagent Ni(NO₃)₂·6H₂O (Ni²⁺ concentration: 6576 mg L⁻¹) with the same Ni²⁺ concentration of the real electroplating wastewater was used for the preparation of a new S_{Fe-Ni} catalyst. Thermocatalytic results shows that the CO rate for S_{Fe-Ni} prepared from real Ni wastewater is the same of that prepared from manmade Ni²⁺ solution (**Figure S10**). This should be attributed to the similar Ni loading for these two samples from the ICP results (**Table S4**). These results suggest that the existence of the other ions besides Ni²⁺ in the real electroplating wastewater has no harm to the upcycling of Ni for the CO₂ catalysis.

A paragraph has been added as the third paragraph in the ‘Thermocatalytic performance’ part as follows: **As table S3 shows, many other ions beside Ni²⁺ exist in the electroplating wastewater. To verify the influence of these ions on the catalytic performance, a synthetic Ni²⁺ solution made**

from reagent $\text{Ni}(\text{NO}_3)_2 \cdot 6\text{H}_2\text{O}$ with only Ni^{2+} (the Ni^{2+} concentration is the same of that for the real electroplating wastewater) was used as the precursor to prepare the $\text{S}_{\text{Fe-Ni}}$ sample. Notably, **Fig. S10** shows that the CO rate for $\text{S}_{\text{Fe-Ni}}$ prepared from the synthetic Ni^{2+} solution is similar to that for $\text{S}_{\text{Fe-Ni}}$ (prepared from real electroplating wastewater). This should be attributed to the similar loading amount of Ni for these two samples and the trace amount of other ions therein (**Table S4 and Table S5**). These results suggest that the existence of the other ions besides Ni^{2+} in the real electroplating wastewater does not deteriorate the upcycling efficiency of Ni for the CO_2 catalysis reported herein.

A new figure has been added to the supporting information as **Fig. S10**.

Figure S10. CO rates of different $\text{S}_{\text{Fe-Ni}}$ samples (testing condition: 6 mg of $\text{S}_{\text{Fe-Ni}}$ was diluted by 24 mg of commercial SiO_2 , $\text{CO}_2/\text{H}_2/\text{N}_2 = 5/5/20$ mL/min, ambient pressure) and commercial SiO_2 (testing condition: 30 mg of commercial SiO_2 , $\text{CO}_2/\text{H}_2/\text{N}_2 = 5/5/20$ mL/min, ambient pressure). Notably, the CO production rate was greatly improved to be ~ 40 $\text{mmol g}_{\text{cat}}^{-1} \text{h}^{-1}$ by changing the space velocity from 20000 to 400000 $\text{mL g}_{\text{cat}}^{-1} \text{h}^{-1}$.

Table S3. The concentration of different elements in the electroplating wastewater.

Elements	Ni	Na	Ca	K	Mn	Mg	Cu	Cr	Al	B	Si	P
Concentration (mg L ⁻¹)	6576	70.99	10.22	3.87	0.93	0.73	0.08	0.20	0.59	320	0.52	0.67

Table S4. The loading amount of Ni and Fe for the different samples.

Sample	Weight ratio of Ni	Weight ratio of Fe
S _{Fe-Ni} (batch #1) from real electroplating waste water ^a	2%	17%
S _{Fe-Ni} (batch #2) from real electroplating waste water ^a	1.9%	16.8%
S _{Fe-Ni} (batch #3) from real electroplating waste water ^a	2.1%	15.9%
S _{Fe-Ni} from synthetic Ni ²⁺ solution	1.7%	17.9%
S _{Fe}	0	23%
S _{Ni}	2.6%	0

^aThree batches of samples were prepared to verify the reproducibility of the upcycling procedure.

Table S5. The loading amount of B, Na and Ca (which were with the highest concentrations in the original electroplating wastewater) for different S_{Fe-Ni} samples.

Sample	Weight ratio of B	Weight ratio of Na	Weight ratio of Ca
S _{Fe-Ni} (batch #1) from real electroplating wastewater	0.23%	0.038%	0.14%
S _{Fe-Ni} (batch #2) from real electroplating wastewater	0.24%	0.036%	0.097%

S_{Fe-Ni} (batch #3) from real electroplating wastewater	0.24%	0.038%	0.13%
S_{Fe-Ni} from synthetic Ni²⁺ solution	0	0.034%	0.21%

Three batches of S_{Fe-Ni} samples from real electroplating wastewater were prepared to verify the reproducibility of the upcycling procedure.

Q2. Rather than qualitative statements on the role of the middle mantle protecting magnetite from acids, the authors should present experimental data of stability of the iron core against corrosion of acids at various concentrations over the time. It is unclear, why the core can be reduced by H₂ and are protected from protons.

Author Reply: As instructed by the reviewer, an acid etching experiment was conducted to demonstrate the acid resistance of the adsorbent used in this work. A sample prepared by coating CNC with only a mesoporous SiO₂ layer (without the middle mantle protecting layer, denoted as CNC@mSiO₂) was used as a control sample to verify the acid resistance of the dense mantle SiO₂ layer in CNC@SiO₂@mSiO₂ (Fig. S3). The similar particle size distributions of CNC@SiO₂@mSiO₂ and CNC@mSiO₂ suggest that the thickness of SiO₂ for these two samples are the same, making the comparison fair. The concentration of Fe in the etching solution for CNC@SiO₂@mSiO₂ is much lower than that for CNC@mSiO₂ under various etching conditions. These results indicate that the dense silica in the middle layer of CNC@SiO₂@mSiO₂ has a strong resistance against acid. These highlighted sentences have been added to the first paragraph in the ‘Characterization of the catalysts’ part.

In contrast, the reduction by H₂ was performed at high temperatures, so the diffusion of H₂ was facilitated.

A new figure has been added to the supporting information as Fig. S3 and the orders of other

figures have been updated.

The synthesis method of CNC@mSiO₂ and the acid etching method have been added to the supplementary notes.

Figure S3. (a) Scanning electron microscope (SEM) image of CNC coated by a dense silica layer, followed by the coating of a mesoporous SiO₂ layer (denoted as CNC@SiO₂@mSiO₂). Scale bar, 500 nm. (b) SEM image of CNC coated by a mesoporous SiO₂ layer (denoted as CNC@mSiO₂). Scale bar, 500 nm. (c) The concentration of Fe in the etching solution. The inserts in (a) and (b) are the particle size distributions of the corresponding samples.

Q3. I do not understand the nomenclature of SFe-Ni. It confuses the chemical reader with the presence of sulfur in the material. In addition, the chemical composition of the catalyst, particularly the Ni content and the presence of other metal impurities should be given.

Author Reply: We thank the reviewer for raising this concern. It should be much clearer to use the chemical composition to name the catalyst. However, the composition of our catalyst is so complicated that the name would be long and complex which is not convenient to use (e.g. the chemical composition of the S_{Fe-Ni} sample is 0.16Fe@SiO₂@mSiO₂-0.02Ni). The ‘S’ in different samples is the abbreviation of the sample. To make the expressions clearer, we have made changes to the nomenclature by switching the main components of a specific sample to the subscript to avoid misinterpreting “S” as an element like Fe and Ni (e.g. The SFe-Ni sample has been changed to be the S_{Fe-Ni} sample).

Q4. The previous point is in connection of how to present the results. I think that giving the CO production as mol/gNi can be misleading. Data should be given, at least, per total mass of the photocatalyst in abstract and other parts of the manuscript.

Author Reply: The activity calculated based on the total mass of the catalyst has been added to the abstract and was already given in other relevant parts.

Q5. Specifically in page 10, line 231 a CO productivity rate of 5 mmol/gcat h for a reaction carried out at 500 °C is not an impressive number and it is in fact below many other thermal catalysts at this high temperature. Even more, the value of 4.87 mmol/gcat h indicated in page 11, line 264 is also low compared to the state of the art. In this part, the surface illuminated by the beam of 190 W must be given to have an idea of the number of suns used in this measurement.

Author Reply: We thank the reviewer for raising this important point. We have updated the photothermal catalytic data by using higher light intensities with the help of light concentrator. The photothermal catalytic activity could be further improved to be 306.7 mmol g_{Ni}⁻¹ h⁻¹ (9.73 mmol g_{cat}⁻¹ h⁻¹) in the flow reactor (**Fig. 4**), while maintaining ~100% selectivity towards CO production. Notably, the photothermal catalytic CO rate for S_{Fe-Ni} could be further improved to be 44.1 mmol·g⁻¹·h⁻¹ (1.9 mol·g_{Ni}⁻¹·h⁻¹) by using a batch reactor.

The number of suns has been included in the new graph.

The photothermal catalytic activity of our $S_{\text{Fe-Ni}}$ catalyst was also compared with those for the commercial Cu-ZnO- Al_2O_3 catalyst and some reported cases (**Fig. S21 and Table S7**). The Cu-ZnO- Al_2O_3 sample shows lower CO rate than $S_{\text{Fe-Ni}}$ under similar testing conditions. Moreover, our $S_{\text{Fe-Ni}}$ catalyst has higher or at least the same order of production rate of those for all the other listed catalysts in **Table S7**.

Figure 4. (a) Diffuse reflectance spectra of different catalysts. (b) CO production rates of different samples illuminated with a Xe arc lamp supplemented with a light concentrator in the flow reactor. The power of the lamp was set to be 190, 200, and 210 W, corresponding to the light intensities of 28.3, 30.7 and 33.3 suns, respectively. (c) Surface temperature profiles of different samples illuminated with concentrated light from a Xe arc lamp. The power of the lamp was set to be 210 W. (d) CO production rates of different samples in the batch reactor illuminated with a Xe arc lamp supplemented with a light concentrator. The light intensity was 28 suns. The CO rates with the unit of ‘ $\text{mmol}\cdot\text{g}_{\text{Ni}}^{-1}\cdot\text{h}^{-1}$ ’ for $S_{\text{Fe-Ni}}$ in (b) and (d) were calculated based on the mass of Ni, and the activity of the isolated Fe component was deducted by subtracting the CO rate of S_{Fe} under the same

testing condition. The red data points in (b) and (d) correspond to the selectivity towards production of CO.

Figure S21. Photothermal catalytic CO rates of Cu-ZnO-Al₂O₃ (testing condition: 1.5 mg of commercial Cu-ZnO-Al₂O₃ was diluted by 28.5 mg of commercial SiO₂, CO₂/H₂/N₂ = 2.5/2.5/5 mL/min, ambient pressure) and S_{Fe-Ni} (testing condition: 30 mg, CO₂/H₂/N₂ = 2.5/2.5/5 mL/min, ambient pressure). The Cu weight ratio of commercial Cu-ZnO-Al₂O₃ was determined by ICP-OES to be 42.9%. The diluted Cu-ZnO-Al₂O₃ sample (2.1 wt% Cu) in the testing condition has similar weight ratio of active metal as S_{Fe-Ni} (2.0 wt% Ni). The samples were illuminated with a Xe arc lamp supplemented with a light concentrator in the flow reactor. The light intensity was set to be 26.1 suns. The CO rates with the unit of 'mmol·g_{metal}⁻¹·h⁻¹' for S_{Fe-Ni} was calculated based on the mass of Ni, and the activity of the isolated Fe component was deducted by subtracting the CO rate of S_{Fe} under the same testing condition.

Table S7. Comparison of the catalytic performance of our catalyst and the other reported photothermal catalysts tested in batch reactors.

Catalyst	Metal	CO ₂ : H ₂	Light source	Light intensity	CO selec.	R _{max}
	[wt%]			[mW/cm ²]		[mol g _{metal} ⁻¹ h ⁻¹]

Ni/Al ₂ O ₃ ⁷	2.1	1:4	300 W Xe light (UV-Vis-NIR)	N/A	0.95%	2.3
Co/Al ₂ O ₃ ⁷	2.5	1:4	300 W Xe light (UV-Vis-NIR)	N/A	0.49%	0.9
Pd/Al ₂ O ₃ ⁷	2	1:4.1	300 W Xe light (UV-Vis-NIR)	N/A	1.36%	0.53
Pt/Al ₂ O ₃ ⁷	2.4	1:4.1	300 W Xe light (UV-Vis-NIR)	N/A	84.85%	0.47
Ir/Al ₂ O ₃ ⁷	2.8	1:4.1	300 W Xe light (UV-Vis-NIR)	N/A	36.74%	0.05
Ni/N _{5.0} -CeO ₂ ⁸	10	1:1	300 W Xe light (UV-Vis-NIR)	2110	~100%	0.209
Ru/TiNT ⁹	11.2	1:4	Newport solar simulator	150 + 210°C	N/A	0.1107
Co@CoN&C-1 ¹⁰	76.3	1:1	300 W Xe light	N/A	91.1%	0.17
Ni/80Ce-20Ti ₂ SG ¹¹	10	1:4	300 W Xe light	N/A	N/A	0.17
Pd@Nb ₂ O ₇ ¹²	0.5	1:1	300 W Xe light	2500	N/A	0.98
S _{Fe-Ni} (this work)	Ni: 2	1:1	300 W Xe light (UV-Vis-NIR)	2800	> 99.8%	1.9*

*The CO rates were calculated based on the mass of Ni, and the activity of the isolated Fe component was deducted by subtracting the CO rate of S_{Fe} under the same testing condition.

Q6. In page 10, lines 226-228, the reader is disoriented about whether or not the experiments have been carried out with a photocatalyst prepared using real electroplating waste water. If this were the case, then, to have reproducible results, full analytical data of the material must be given.

Author Reply: All characterization and performance tests if not otherwise specified were on samples prepared using real electroplating water. We have reiterated this in the revised manuscript to avoid confusion. Furthermore, another two batches of the $S_{\text{Fe-Ni}}$ samples were prepared from real electroplating wastewater to verify the reproducibility of the upcycling procedure. The loading amounts of the main elements (Fe and Ni) were added to **Table S4**. Although there exist some other ions besides Ni^{2+} in the wastewater, the concentrations of most of these ions are too low to be considered. Only the concentrations of B, Na and Ca ions are relatively higher than the other ions besides Ni in the wastewater. Three batches of $S_{\text{Fe-Ni}}$ samples prepared from real wastewater and a $S_{\text{Fe-Ni}}$ sample from a synthetic Ni^{2+} solution were dissolved and the loading amount of B, Na and Ca ions in these samples were detected (**Table S5**). The almost same loading amounts of B, Na and Ca in the three batches of the $S_{\text{Fe-Ni}}$ samples from real electroplating wastewater demonstrate the reproducibility of the upcycling procedure. Moreover, the $S_{\text{Fe-Ni}}$ sample prepared from the synthetic Ni^{2+} solution contains no B elements, again demonstrating that our catalysts were prepared from real electroplating wastewater. It is noticed that both the $S_{\text{Fe-Ni}}$ samples from wastewater and a synthetic Ni^{2+} solution contain negligible amounts of Na and Ca, which might be attributed to the existence of trace amounts of Na and Ca in the reagents or acid solution used to dissolve the samples. This paragraph has been added to the supporting information below **Table S5**.

A table has been added to the supporting information as **Table S5**.

A sentence has been added to the first paragraph in the ‘Preparation of the catalysts’ part as follows: A control sample was prepared by replacing the electroplating wastewater with a synthetic Ni^{2+} solution from reagent $\text{Ni}(\text{NO}_3)_2 \cdot 6\text{H}_2\text{O}$. The compositions of the prepared catalysts are listed in **Table S4** and **Table S5**.

Table S5. The loading amount of B, Na and Ca (which were with the highest concentrations in the original electroplating wastewater) for different $S_{\text{Fe-Ni}}$ samples.

Sample	Weight ratio of B	Weight ratio of Na	Weight ratio of Ca
$S_{\text{Fe-Ni}}$ (batch #1) from real electroplating wastewater	0.23%	0.038%	0.14%
$S_{\text{Fe-Ni}}$ (batch #2) from real electroplating wastewater	0.24%	0.036%	0.097%
$S_{\text{Fe-Ni}}$ (batch #3) from real electroplating wastewater	0.24%	0.038%	0.13%
$S_{\text{Fe-Ni}}$ from manmade Ni^{2+} solution	0	0.034%	0.21%

Three batches of $S_{\text{Fe-Ni}}$ samples from real electroplating wastewater were prepared to verify the reproducibility of the upcycling procedure.

Q7. I do not agree that bed configuration is the reason why the flow reaction gives higher CO production, The authors should consider the influence of H₂O in the batch reaction, poisoning the catalyst, while in flow, the H₂O concentration could be steady during the process. Since the negative influence of H₂O on the photothermal reaction has been reported, the authors have to perform measurements in the presence of controlled amounts of H₂O in the feed and determine the influence of the presence of this product.

Author Reply: The reviewer might have misread the catalytic results. The catalytic activity in the batch reactor is higher rather than lower than that in the flow reactor.

Q8. The section of cycle experiments is less convincing, since the results could vary depending in the real source. There are currently various electroplating systems with various compositions. However, from the catalysis point of view, the worrying point is that over the time, the composition of the catalyst will vary due to the different percentages of adsorbed metals. Here in the text, only Ni adsorption is commented, but the other metal components will also change in proportion upon cycle. I am not thinking that the concept could not work, I am only saying that the situation is far more complex than presented in this simple case.

Author Reply: We thank the reviewer for raising these thought-provoking points. The composition of the wastewater is complex and the adsorption conditions in different cycles are different. Nevertheless, our previous results could at least demonstrate that the catalytic activity could be retained for nine cycles after which most of the Ni (78%) in the wastewater could be recycled. In the later cycles, although the product rate decreased, the CO selectivity always keeps at ~100% (Fig. S24).

As the following table shows, the loading amounts of B, Na, and Ca for S_{Fe-Ni} in the 9th and 15th cycle were detected. Trace amounts of these ions were found on these two samples. These results further eliminate the side effect of the other ions besides Ni²⁺ in the electroplating wastewater.

The loading amount of B, Na and Ca for different S_{Fe-Ni} samples in the cycle experiment.

Sample	Weight ratio of B	Weight ratio of Na	Weight ratio of Ca
S _{Fe-Ni} after 9 cycles in the cycle experiment	0.053%	0.03%	0.34%
S _{Fe-Ni} after 15 cycles in the cycle experiment	0.044%	0.048%	0.4%

Response to Reviewer #3:

General Comments: In this article, real waste Ni from the electroplating industry was demonstrated as an effective resource for the advanced technology of photothermal CO₂ catalysis. The authors bridged the magnetic adsorption technique with photothermal catalyst design, and constructed a planet earth-mimic structure which indeed has the “greenhouse effect” that is beneficial for photothermal CO₂ catalysis. This is a smart study and it targets multiple hot issues which will draw broad interest, so I recommend publication after addressing my queries:

Q1. The authors have employed mesoporous SiO₂. What is the rationale besides the large surface area? Is the small size of Ni attributed to the confinement of the mesopores?

Author Reply: We thank the reviewer for making this thought-provoking point.

There should be no other functions of the mesoporous SiO₂ used here besides the enlarged surface area. The small size of Ni could likely be attributed to the anchoring effect of the amino groups. A control experiment has been conducted to justify this viewpoint.

To examine whether the mesopores contribute to the small size of Ni and in turn the great selectivity, another control sample was prepared by using solid SiO₂ spheres as the support to graft amino groups, followed by the adsorption and activation of Ni similar to other samples (denoted as S_{SiO₂-Ni}). As Fig. S9a-b shows, no large Ni particles and no obvious crystalline Ni signals could be observed from the TEM and XRD results, respectively. Moreover, the thermocatalytic results suggest that CO was the only product for S_{SiO₂-Ni} which is characteristic of small-sized Ni (Fig. S9c). The BET surface area of amino grafted solid SiO₂ (S_{SiO₂-NH₂}) is much lower than that of CNC@SiO₂@mSiO₂-NH₂ (15.9 m²/g vs. 301.7 m²/g, Fig. S9d). This leads to the much poorer catalytic activity of S_{SiO₂-Ni}, only half of that of S_{Fe-Ni}. These results indicate that the mesoporous SiO₂ only act as an effective support with high specific surface area. This paragraph has been added as the second paragraph in the ‘Thermocatalytic performance’ part.

A figure has been added to the supporting information as Fig. S9. The orders of other figures have been updated as well.

The synthesis methods of S_{SiO₂}, S_{SiO₂-NH₂} and S_{SiO₂-NH₂} have been added to the method part.

Figure S9. (a) TEM image of S_{SiO_2-Ni} . Scale bar, 100 nm. (b) XRD patterns of S_{SiO_2-Ni} and the standard Ni (JCPDS 04-0850). (c) CO rates of S_{SiO_2-Ni} (testing condition: 6 mg of S_{SiO_2-Ni} was diluted by 24 mg of commercial SiO_2 , $CO_2/H_2/N_2 = 5/5/20$ mL/min, ambient pressure) and commercial SiO_2 (testing condition: 30 mg of commercial SiO_2 , $CO_2/H_2/N_2 = 5/5/20$ mL/min, ambient pressure). (d) N_2 adsorption-desorption isotherms of $CNC@SiO_2@mSiO_2-NH_2$ and amino grafted S_{SiO_2} ($S_{SiO_2-NH_2}$). Notably, the CO production rate was greatly improved to be ~ 40 mmol $g_{cat}^{-1} h^{-1}$ by changing the space velocity from 20000 to 400000 mL $g_{cat}^{-1} h^{-1}$.

Q2. The authors mentioned that more Ni could be seen near the outer part but no serious aggregation was found, after the long-term test. Although the TEM image shows some evidence, XRD result after the test should be provided to prove that the Ni was still highly dispersed without forming nanoparticles.

Author Reply: We thank the reviewer for this valuable suggestion. The XRD patterns of the used samples have been collected and added to the supporting information as **Figure S12**. No signals of crystalline Ni were observed for $S_{\text{Fe-Ni}}$ (used) and S_{Ni} (used).

A sentence has been added to the fourth graph in the ‘Thermal catalytic performance’ part as follows: **The high dispersion of Ni of the used samples could also be evidenced by the XRD patterns which show no characteristic signals of crystalline Ni (Fig. S12).**

Figure S12. XRD patterns of the different samples and the standard Fe (JCPDS 06-0696). The used $S_{\text{Fe-Ni}}$, S_{Fe} , and S_{Ni} were denoted as $S_{\text{Fe-Ni}}$ (after), S_{Fe} (after), and S_{Ni} (after), respectively.

Q3. Why was Ni slightly oxidized after the photothermal test?

Author Reply: The existence of oxidizing CO₂ might lead to the partially oxidation of the highly active Ni atoms at high temperatures.

Q4. Can this upcycling approach be extended to treating other kinds of wastewater, e.g. containing Cu cations?

Author Reply: This is an inspiring question. This upcycling approach might be extended to some other systems if the other pollutant ions besides the major and target waste ion has tiny influence on the catalytic performance.

As a quick proof-of-concept, a synthetic Cu²⁺ solution (Cu²⁺ concentration: 7000 mg/L) prepared from Cu(NO₃)₂·3H₂O was utilized as the Cu precursor to prepare a CNC@SiO₂@mSiO₂-NH₂ loaded Cu sample (denoted as S_{Fe-Cu}). Shown by the graph below, the CO rate for S_{Fe-Cu} is even lower than the calcined adsorbent (S_{Fe}) without capture of any metals. Therefore, this specific upcycling procedure for Ni towards CO₂ catalysis might not be suitable for this Cu wastewater system which is obviously different. However, this result proves the main point in this manuscript that Ni electroplating wastewater stands out as a promising system for upcycling towards catalysts.

Nevertheless, we are endeavoring in developing specific technologies for upcycling other waste materials, and further optimization of the adsorbing materials to enhance its capacity.

Q5. Nickel has showed a wide range of applications in photothermal catalysis. The authors can add some references of nickel in introduction, such as Nat. Commun. 12, 3996, (2021), Nat. Commun. 10, 2359, (2019).

Author Reply: The second reference is indeed relevant and insightful, thus has been added to the introduction part as ref [27].

REVIEWER COMMENTS

Reviewer #1 (Remarks to the Author):

The authors have addressed all the comments raised. The manuscript is now acceptable.

Reviewer #2 (Remarks to the Author):

The authors have taken my previous comments in a constructive way and have made changes addressing my remarks. There are, however, some final points that need to be further tackled. The authors may consider the following issues:

I still think that, although the concept of upcycling a waste is very important and an advance in the direction of a sustainable society, the problems associated to it should be clearly presented. In the introduction I am missing a critical comment on how impurities, even in ppm, affect the performance of catalysts. In this regard, the authors have shown that here this is not the case. I see, nevertheless, that the so-called man-made SFe-Ni sample exhibits a bit higher activity than the SFe-Ni made using electroplating waste water and this could deserve a comment. Similarly, the fact that the Ni adsorption on the mesoporous silica decreases upon continued catalyst preparation and for the fifteenth sample the Ni content is one half should clearly impact the photocatalytic performance. I was hoping a critical comment also on this. Overall I think that the upcycling comment has been too enthusiastically presented without a clear discussion of the associated problems. These problems are not a demerit of the upcycling, but challenges to be overcome.

When commenting on the three samples prepared, it has been clearly indicated if they were prepared from the same wastewater sample or from three randomly chosen wastewaters sampled in different days/conditions.

Figure 4b showing the influence of the light intensity should be completed by adding the temperature of the measurement.

The use of only 1.5 mg of Cu-ZnO/Al₂O₃ vs. 30 mg SFe-Ni can make the comparison unfair due to the much high metal loading. If 1.5 mg of Cu-Zn/Al₂O₃ is used to keep similar the metal loading, then 28.5 mg of the support should also be mixed to improve dispersion and increase the surface area exposed.

After these final comments are solved, publication is recommended.

Reviewer #3 (Remarks to the Author):

The authors have addressed all my comments for this paper. The paper has been significantly improved after revising, so I recommend accepting this paper for publication in Nature Communications.

Response to Reviewer #2:

Q1. In the introduction I am missing a critical comment on how impurities, even in ppm, affect the performance of catalysts. In this regard, the authors have shown that here this is not the case.

Author Reply: These are really valuable suggestions. We totally agree that impurities could significantly affect the activity of catalysts in some situations. For example, Mikhail *et al*'s work (*Fuel* 2021, 306, 121639) suggests that the presence of Na and K impurities in Ni/CeZrO_x catalysts resulted in decreased CO₂ conversions, lower selectivity to CH₄ and increased power consumption in DBD plasma-catalytic CO₂ methanation.

Alternatively, under many conditions, impurities could instead act as promoters in catalysis. Tasfy *et al.* demonstrated that Pb and Mn are promoters for Cu/ZnO-catalyst in CO₂ hydrogenation to methanol (*PROCESS AND ADVANCED MATERIALS ENGINEERING* Vol. 625, 2014, pp. 289.). The introduction of alkali metals (e.g. Na and K) were also found to improve the catalytic performance in some cases (*J Environ Chem Eng* 2016, 4, 2725–35. *Appl Catal B: Environ* 2018, 236, 162. *J. Catal.* 1985, 93, 152.).

Therefore, the impurities in the wastewater might work as either a promoter or an inhibitor. That is to say, not all the wastewaters can be utilized for the upcycling procedure described herein. The selection of suitable wastewater systems and catalytic reactions are thus critical, and the resulting activity must be explored through well-planned systematic experimentation.

We have added the following sentences to the end of the second paragraph in the introduction:

For example, Mikhail *et al*'s work suggests that the presence of Na and K impurities in Ni/CeZrO_x catalysts resulted in decreased CO₂ conversions, lower selectivity to CH₄ and increased power consumption in DBD plasma-catalytic CO₂ methanation.³¹ In contrast, the impurities can also function as promoters rather than inhibitors. In many cases, different from Maria's case, the introduction of alkali metals (e.g., Na and K) could instead improve the catalytic performance.³²
³⁴ Nevertheless, the existence of impurities in wastewater still brings complications and uncertainties, so judicious design of the upcycling process, materials, and catalysts are critical and must be validated.

Q2. The so-called man-made SFe-Ni sample exhibits a bit higher activity than the SFe-Ni made using electroplating waste water and this could deserve a comment.

Author Reply: The slightly lower CO rate of $S_{\text{Fe-Ni}}$ prepared from real electroplating wastewater rather than $S_{\text{Fe-Ni}}$ prepared from the synthetic Ni^{2+} solution at 500 °C might be ascribed to the slightly higher Fe loading for the latter (17.9 wt%) than the former (15.9 wt%) determined by ICP-OES. As Fig. S7 shows, the Fe component exhibits a significant contribution to the production rate at 500 °C in thermocatalytic tests. While at 400 °C, this contribution is much smaller. That might be the reason for the same CO rate of these two kinds of $S_{\text{Fe-Ni}}$ samples at 400 °C. This paragraph has been added to the description of Fig. S10 in the supporting information.

We have updated the descriptions (the third sentence of the third paragraph in the ‘Thermocatalytic performance’ part) in the main text as follows: Fig. S10 shows that the CO rate for $S_{\text{Fe-Ni}}$ prepared from real electroplating wastewater is the same of that for $S_{\text{Fe-Ni}}$ prepared from the synthetic Ni^{2+} solution at 400 °C, while only a slightly lower CO rate was found for the former than the latter at 500 °C. Nevertheless, the overall discrepancy between these two $S_{\text{Fe-Ni}}$ samples is minimal.

Figure S7. Thermocatalytic performance of CO₂ hydrogenation. (a) CO rate of different samples under various temperatures. **(b)** CO rate of $S_{\text{Fe-Ni}}$ under 500 °C during 38.7-hour testing. The red data points in (a) correspond to the selectivity towards production of CO.

Q3. The fact that the Ni adsorption on the mesoporous silica decreases upon continued catalyst preparation and for the fifteenth sample the Ni content is one half should clearly impact the photocatalytic performance. I was hoping a critical comment also on this. Overall I think that the upcycling comment has been too enthusiastically presented without a clear discussion of the associated problems. These problems are not a demerit of the upcycling, but challenges to be overcome.

Author Reply: We thank the reviewer for these insightful suggestions. We agree that the drop of activity for the samples in the later cycles due to the lower loading of Ni might be a problem. We have clarified this in the revised manuscript as the reviewer suggested. Moreover, we further propose a possible remedial strategy regarding this problem: performing adsorption in fresh wastewater again using the sample in the later cycles to achieve improved Ni loading. As a simple demonstration, we prepared a $S_{\text{Fe-Ni}}$ sample using diluted Ni wastewater (1 mL of wastewater diluted by 29 mL of milli-Q water) to simulate $S_{\text{Fe-Ni}}$ in the later cycles. Next, a second adsorption process in the fresh Ni wastewater was operated on this sample. The CO rate of the treated sample achieved was $5.3 \text{ mmol g}_{\text{cat}}^{-1} \text{ h}^{-1}$ which is similar to that of $S_{\text{Fe-Ni}}$ prepared using fresh Ni wastewater under the same testing condition, demonstrating the feasibility of the remedial strategy. Nevertheless, second adsorption also made the upcycling procedure more time-consuming. Simpler and more cost-effective upcycling strategies should be developed in future works.

This paragraph has been added as the second paragraph in the 'cycle experiment' part.

Q4. When commenting on the three samples prepared, it has been clearly indicated if they were prepared from the same wastewater sample or from three randomly chosen wastewaters sampled in different days/conditions.

Author Reply: The three samples were prepared from the same wastewater. We have clarified this information in different parts of the updated manuscript.

Q5. Figure 4b showing the influence of the light intensity should be completed by adding the temperature of the measurement.

Author Reply: Three data points were collected to obtain the average CO rate after 10-minutes illumination for each sample. The equilibrium temperatures under illumination for $S_{\text{Fe-Ni}}$, S_{Fe} , S_{Ni} and without catalyst were $401.8 \text{ }^{\circ}\text{C}$, $389.6 \text{ }^{\circ}\text{C}$, $295.3 \text{ }^{\circ}\text{C}$, and $180.8 \text{ }^{\circ}\text{C}$, respectively. These two sentences have been added to the figure caption in the updated manuscript.

Figure 4. (a) Diffuse reflectance spectra of different catalysts. (b) CO production rates of different samples illuminated with a Xe arc lamp supplemented with a light concentrator in the flow reactor. The power of the lamp was set to be 190, 200, and 210 W, corresponding to the light intensities of 28.3, 30.7 and 33.3 suns, respectively. Three data points were collected to obtain the average CO rate after 10-minutes illumination for each sample. (c) Surface temperature profiles of different samples illuminated with concentrated light from a Xe arc lamp. The power of the lamp was set to be 210 W. The equilibrium temperatures under illumination for S_{Fe-Ni} , S_{Fe} , S_{Ni} and without catalyst were 401.8 $^{\circ}C$, 389.6 $^{\circ}C$, 295.3 $^{\circ}C$, and 180.8 $^{\circ}C$, respectively. (d) CO production rates of different samples in the batch reactor illuminated with a Xe arc lamp supplemented with a light concentrator. The light intensity was 28 suns. The CO rates with the unit of ' $mmol \cdot g_{Ni}^{-1} \cdot h^{-1}$ ' for S_{Fe-Ni} in (b) and (d) were calculated based on the mass of Ni, and the activity of the isolated Fe component was deducted by subtracting the CO rate of S_{Fe} under the same testing condition. The red data points in (b) and (d) correspond to the selectivity towards production of CO.

Q6. The use of only 1.5 mg of Cu-ZnO/Al₂O₃ vs. 30 mg S_{Fe-Ni} can make the comparison unfair due to the much high metal loading. If 1.5 mg of Cu-Zn/Al₂O₃ is used to keep similar the metal loading, then 28.5 mg of the support should also be mixed to improve dispersion and increase the surface area exposed.

Author Reply: We agree with the reviewer regarding this point. The Cu-ZnO/Al₂O₃ (1.5 mg) was indeed mixed with 28.5 mg of the support to improve the dispersion in the previous test. We had also clarified the testing condition in the figure caption of **Fig. S21** in the previous manuscript.

Figure S21. Photothermal catalytic CO production rates of Cu-ZnO-Al₂O₃ (testing condition: 1.5 mg of commercial Cu-ZnO-Al₂O₃ was diluted by 28.5 mg of commercial SiO₂, CO₂/H₂/N₂ = 2.5/2.5/5 mL/min, ambient pressure) and S_{Fe-Ni} (testing condition: 30 mg, CO₂/H₂/N₂ = 2.5/2.5/5 mL/min, ambient pressure). The Cu weight ratio of commercial Cu-ZnO-Al₂O₃ was determined by ICP-OES to be 42.9%. The diluted Cu-ZnO-Al₂O₃ sample (2.1 wt% Cu) in the testing condition has similar weight ratio of active metal as S_{Fe-Ni} (2.0 wt% Ni). The samples were illuminated with a Xe arc lamp supplemented with a light concentrator in the flow reactor. The power of the lamp was set to be 180 W, corresponding to the light intensity of 26.1 suns. The CO rates with the unit of 'mmol·g_{metal}⁻¹·h⁻¹' for S_{Fe-Ni} was calculated based on the mass of Ni, and the activity of the isolated Fe component was deducted by subtracting the CO rate of S_{Fe} under the same testing condition.

REVIEWERS' COMMENTS

Reviewer #2 (Remarks to the Author):

The authors have satisfactorily solved my last concerns. Publication in its present form is now recommended.

Response to Reviewer #2:

General Comments: The authors have satisfactorily solved my last concerns. Publication in its present form is now recommended.

Author Reply: We thank the reviewer for the positive appraisal of the revised work.